# What Does Preference Learning Recover from Pairwise Comparison Data?

**Rattana Pukdee** [1]   **Nina Balcan** [1]   **Pradeep Ravikumar** [1]

## Abstract

Pairwise preference learning is central to machine learning, with recent applications in aligning language models with human preferences. A typical dataset consists of triplets $(x, y^+, y^-)$, where response $y^+$ is preferred over response $y^-$ for context $x$. The Bradley–Terry (BT) model is the predominant approach, modeling preference probabilities as a function of latent score differences. Standard practice assumes data follows this model and learns the latent scores accordingly. However, real data may violate this assumption, and it remains unclear what BT learning recovers in such cases. Starting from triplet comparison data, we formalize the preference information it encodes through the *conditional preference distribution* (CPRD). We give precise conditions for when BT is appropriate for modeling the CPRD, and identify factors governing sample efficiency—namely, margin and connectivity. Together, these results offer a data-centric foundation for understanding what preference learning actually recovers.

## 1. Introduction

Learning from pairwise comparisons is a classical learning paradigm in machine learning. in this setting, the supervision is usually in the form of triplet

$$(x, y^+, y^-), \tag{1}$$

indicating that response $y^+$ is preferred over $y^-$ in context $x$. This mode of supervision is attractive because it is often far easier for humans to express preferences than to assign calibrated scores. Our goal is then to learn the preference information from this triplet data where when we observe a new data point $x$ and a pair of responses $\{y, y'\}$, we can accurately predict if $y$ is more preferable than $y'$ or not.

[1] Machine Learning Department, Carnegie Mellon University, Pittsburgh, USA. Correspondence to: Rattana Pukdee <rpukdee@cs.cmu.edu>.

*Proceedings of the 43rd International Conference on Machine Learning*, Seoul, South Korea. PMLR 306, 2026. Copyright 2026 by the author(s).

Due to the ease of data collection, this learning paradigm has various applications from sport team ranking (Bradley & Terry, 1952; Bozóki et al., 2016), recommendation systems (Fürnkranz & Hüllermeier, 2010; Qian et al., 2015; Osadchiy et al., 2019), psychology (Thurstone, 1927; Luce et al., 1959; David, 1963), learning combinatorial valuations (Balcan et al., 2016) and recently, large language models alignment to human preferences (Christiano et al., 2017; Ouyang et al., 2022; Bai et al., 2022b;a; Rafailov et al., 2023).

The dominant approach is to fit a Bradley–Terry model (BT) to the observed comparisons (Bradley & Terry, 1952; Luce et al., 1959). This model posits that each item has a latent quality score, and that pairwise preferences are generated stochastically according to these scores. BT has been studied extensively from a statistical perspective, with well-established results on consistency and sample complexity for recovering these latent scores. Such analyses assume that true scores exist in the first place. However, comparison data arises from diverse sources, from human annotators to AI labeling systems for which they may not necessarily arise from the generative process that involves latent scores. Samples from the triplet distribution are what we observe and this suggests the triplet distribution is the natural starting point. Rather than assume BT, we begin with the data and ask:

- What preference information is actually encoded by a triplet distribution ?

- When can these preferences be represented by a BT model?

- What does BT learning recover when the model is misspecified?

- How do data collection choices affect learnability and sample efficiency?

Our main contribution is a clean foundation for preference learning from comparison data. With precise answers to these questions, practitioners can reason about data collection, understand what their models learn, and diagnose failures when assumptions do not hold.

**Contributions.** This paper develops a framework that makes these questions precise and provides clear answers:

1. **Conditional Preference Distribution (CPRD).** We formalize the preference information encoded in any triplet distribution $P$ through the CPRD, the probability of preferring $y$ over $y'$ given context $x$. This is the fundamental object encoded in comparison data, independent of how it was generated.

2. **BT Characterization.** We characterize the relationship between BT representability and *positive–negative conditional independence* (given the context, the winning and losing responses are generated independently of each other) and we show that the two are equivalent up to the choice of triplet distribution. This provides a precise condition for when BT modeling is appropriate.

3. **Learning Objective Interpretation.** We prove that the BT learning objective is equivalent to a KL projection of the CPRD to the BT family at the *preference level*. This clarifies what BT learning optimizes and what it recovers when the model is misspecified.

4. **Sample Complexity Analysis.** We show that two factors govern learning efficiency: *pairwise margin* and *comparison connectivity*. We provide estimation error bounds that make these dependencies precise, offering guidance for data collection. We validate these findings empirically.

By starting from the data distribution, we unify known and new results — giving practitioners a coherent framework for understanding what preference learning recovers and when to trust it.

## 2. Related Work

**Bradley–Terry and representability.** The Conditional Pairwise Preference Distribution (CPRD) corresponds to what is referred to as a choice probability in classical paired-comparison models (Thurstone, 1927; Bradley & Terry, 1952; Luce et al., 1959; Plackett, 1975), ranking from pairwise comparisons (Hüllermeier et al., 2008; Fürnkranz & Hüllermeier, 2010; Shah et al., 2016; Heckel et al., 2018; 2019), and dueling bandits (Yue et al., 2012; Dudík et al., 2015; Jamieson et al., 2015; Sui et al., 2018). The Bradley–Terry model is a popular approach for learning preferences from such data; see (Hamilton et al., 2023) for a comprehensive review. Prior work has characterized when a distribution can be represented by the BT model (Good, 1955; Luce et al., 1959; Buhlmann & Huber, 1963; Yellott, 1977; Falmagne, 1978; Breitmoser, 2021), with transitivity being a well-known limitation that several methods attempt to address (Chen & Joachims, 2016; Bower & Balzano, 2020;

Tatli et al., 2024; Zhang et al., 2025a). In contrast, we study BT representability starting from a given triplet distribution rather than assuming data follows the BT model.

**Statistical analysis.** A large body of work has analyzed the consistency of maximum likelihood estimation for the Bradley–Terry model, showing that MLE recovers the true parameters when data follows the BT model (Hunter, 2004; Huang et al., 2006; Yan et al., 2012; Han et al., 2020; Bong & Rinaldo, 2022; Gao et al., 2023a). Many works provide sample complexity bounds for parameter recovery (Negahban et al., 2012; Hajek et al., 2014; Chen & Suh, 2015; Shah et al., 2016; Bong & Rinaldo, 2022; Li et al., 2022), with extensions to parametric settings where scores are linear (Zhu et al., 2023; Fan et al., 2024) or parameterized by neural networks (Sun et al., 2025). We show that under positive–negative conditional independence, BT recovers the log density ratio between the positive and negative distributions. This quantity also appears in noise contrastive estimation (NCE), where one distinguishes data from noise (Gutmann & Hyvärinen, 2010; 2012; Mnih & Kavukcuoglu, 2013); the related InfoNCE loss has become widely used in representation learning (Oord et al., 2018; Chen et al., 2020; Wang & Isola, 2020). Prior work has proposed metrics for evaluating preference dataset quality, including information content based on embedding similarity (Shen et al., 2024) and truncated influence functions (Zhang et al., 2025b). More broadly, choosing which comparisons to collect for efficient preference recovery relates to experimental design (Graßhoff & Schwabe, 2008; Guo et al., 2018) and active learning (Jamieson & Nowak, 2011; Houlsby et al., 2011; Heckel et al., 2019; Das et al., 2025; Shen et al., 2025).

**Preference learning for large language models.** Preference learning from pairwise comparisons is central to aligning language models with human preferences (Christiano et al., 2017; Ouyang et al., 2022; Bai et al., 2022b;a). The standard approach learns a reward model from pairwise comparisons, then uses it to align the language model via reinforcement learning. Recent work also aligns the language models directly from pairwise comparisons (Rafailov et al., 2023; Zhu et al., 2023; Ethayarajh et al., 2024; Li et al., 2024; Chen et al., 2024). This paper focuses on the reward learning step: characterizing the reward learned from a given dataset. A separate question is what makes a good reward model: ideally, high reward should correspond to good downstream performance. However, reward hacking, where the model achieves high scores without improving on downstream tasks, remains a challenge (Skalse et al., 2022; Gao et al., 2023b). Several benchmarks evaluate reward model quality (Lambert et al., 2025; Liu et al., 2025b; Zhou et al., 2025), and theoretical work has sought to characterize what makes a good reward (Razin et al., 2025). Recent findings suggest that rewards with high variance are more

effective at teaching an LLM (Razin et al., 2025; Xu et al., 2025; Guo et al., 2025). Our results provide a complementary perspective: a target reward with large margin is also easier to learn.

## 3. Setup

Let $\mathcal{X}$ be a context space and $\mathcal{Y}$ be a set of outcomes. We are interested in learning which outcome $y$ is preferable for a given context $x$. We focus on learning this preference from pairwise comparisons. Formally, we assume that there exists a joint distribution $P$ over $\mathcal{X} \times \mathcal{Y} \times \mathcal{Y}$ where each triplet $(x, y^+, y^-) \sim P$ represents that for a given context $x$, an outcome $y^+$ is preferable to an outcome $y^-$. We use $\succ$ to denote the preference relation, i.e., $y \succ y'$ means that $y$ is preferable to $y'$. For convenience, we write $P(x, y, y') := P(x, y^+ = y, y^- = y')$ for the probability (or density) of the oriented triple $(x, y, y')$.

First, we observe that the distribution $P$ over triplets induces a preference distribution over any pairs of outcomes for a given context. We formalize this as the conditional preference distribution (CPRD) which is defined as follows:

**Definition 3.1** (Conditional Preference Distribution (CPRD))**.** For a distribution $P$ over $\mathcal{X} \times \mathcal{Y} \times \mathcal{Y}$, the conditional preference distribution (CPRD) of $P$ is defined on any unordered pair $\{y, y'\}$ with $y \neq y'$ and $x \in \mathcal{X}$ as follows:

$$\omega_P(y \succ y' \mid x) = P(y \succ y' \mid x, \{y, y'\}). \quad (2)$$

In fact, by Bayes' rule, we have

$$\omega_P(y \succ y' \mid x) = \frac{P(x, y, y')}{P(x, y, y') + P(x, y', y)}. \quad (3)$$

CPRD directly captures the preference structure of the distribution $P$ and is the target distribution that we aim to learn.

Next, we introduce the Bradley–Terry model which is a popular model for preference learning from pairwise comparisons.

**Definition 3.2** (Bradley–Terry Model)**.** Bradley–Terry model assumes that there exists a score function $r : \mathcal{X} \times \mathcal{Y} \to \mathbb{R}$ such that for any context $x \in \mathcal{X}$ and any unordered pair $\{y, y'\} \in \mathcal{Y}$, the probability of preferring $y$ over $y'$ is given by:

$$P(y \succ y' \mid x) = \sigma(r(x, y) - r(x, y')) \quad (4)$$

where $\sigma(t) = \frac{1}{1+e^{-t}}$ is the sigmoid function.

In practice, one often fits this Bradley–Terry model by minimizing the negative log-likelihood of the observed triplets $(x, y^+, y^-) \sim P$. Concretely, for a set of triplets

$S = \{(x_i, y_i^+, y_i^-)\}_{i=1}^n$, the empirical risk is given by:

$$\widehat{\mathcal{L}}_{\mathrm{BT}}(r) = \sum_{i=1}^n \log \sigma(r(x_i, y_i^+) - r(x_i, y_i^-)). \quad (5)$$

and the learned score function is given by $\hat{r} \in \arg\min_r \widehat{\mathcal{L}}_{\mathrm{BT}}(r)$. We can use $\hat{r}$ to predict the preference between any two outcomes for a given context using equation (4). We remark that the score function $r$ is not unique and is determined up to a constant shift since adding any constant to $r$ still preserves the difference $r(x, y) - r(x, y')$ for any $x$ and $y \neq y'$.

While this gives us a convenient way to learn the preference, it is not clear whether the Bradley–Terry model is able to recover the correct CPRD of the distribution $P$ since the assumption on the structure of the preference (equation (4)) may not hold. Our first contribution is to identify when a CPRD can be represented with a Bradley–Terry model.

## 4. When does a CPRD admit a Bradley–Terry model?

In this section, we characterize when a CPRD is representable by a BT model.

**Definition 4.1.** A CPRD $\omega_P$ is representable by a Bradley–Terry model if there exists a score function $r : \mathcal{X} \times \mathcal{Y} \to \mathbb{R}$ such that for all $x$ and all $y \neq y'$ with $P(x, y, y') + P(x, y', y) > 0$,

$$\omega_P(y \succ y' \mid x) = \sigma(r(x, y) - r(x, y')). \quad (6)$$

We can show that a CPRD is representable by a BT model when the ratio of the probability has a certain structure.

**Proposition 4.2** (CPRD BT factorization)**.** *For a distribution $P$ over $\mathcal{X} \times \mathcal{Y} \times \mathcal{Y}$, the CPRD $\omega_P$ is representable by a BT model if and only if there exists a strictly positive function $h : \mathcal{X} \times \mathcal{Y} \to \mathbb{R}$ such that for all $x$ and all $y \neq y'$ with $P(x, y, y') + P(x, y', y) > 0$,*

$$\frac{P(x, y, y')}{P(x, y', y)} = \frac{h(x, y)}{h(x, y')}. \quad (7)$$

*Proof in Appendix A.*

This proposition shows that not every $P$ induces a CPRD that is representable by a BT model; BT representability requires the probability ratios to have a specific structure. While the condition involves the ratio of $P(x, y, y')$ and $P(x, y', y)$, which seems to be difficult to verify in practice, we will show that data generating process does provide a nice alternative view. First, we introduce a positive–negative conditional independence assumption.

**Definition 4.3** (Positive–negative conditional independence)**.** A distribution $P$ is said to satisfy positive–negative

conditional independence if there exist a marginal $P_X$ on $\mathcal{X}$ and conditional distributions $p_+(\cdot \mid x)$ and $p_-(\cdot \mid x)$ on $\mathcal{Y}$ such that

$$P(x, y^+, y^-) = P_X(x)\, p_+(y^+ \mid x)\, p_-(y^- \mid x). \quad (8)$$

Equivalently, to generate a triplet $(x, y^+, y^-) \sim P$, we first sample $x \sim P_X$ and then we sample $y^+ \sim p_+(y^+ \mid x)$ and $y^- \sim p_-(y^- \mid x)$ independently.

**Theorem 4.4** (CPRD of conditionally independent distribution and BT model)**.** *Let $P$ be a distribution over $\mathcal{X} \times \mathcal{Y} \times \mathcal{Y}$, then the following holds;*

1. *If $P$ satisfies positive–negative conditional independence, then the CPRD $\omega_P$ is representable by a BT model.*

2. *If the CPRD $\omega_P$ is representable by a BT model, then there exists a positive–negative conditional independence distribution $Q$ such that $\omega_Q = \omega_P$.*

*In addition, the suitable score function $r$ for a positive–negative conditional independence distribution $P$ is given by*

$$r(x, y) = \log \frac{p_+(y \mid x)}{p_-(y \mid x)}. \quad (9)$$

*Proof in Appendix A.*

This generative perspective yields a clean sufficient route to BT: if $P$ is generated by independently sampling $y^+ \sim p_+(\cdot \mid x)$ and $y^- \sim p_-(\cdot \mid x)$ given $x$, then $\omega_P$ is automatically representable by a BT model, with score $r(x, y) = \log \frac{p_+(y|x)}{p_-(y|x)}$. This ratio has appeared in the noise contrastive estimation framework (Gutmann & Hyvärinen, 2010) and also resembles the implicit reward function (Rafailov et al., 2023). Importantly, this assumption is not merely descriptive, in many modern settings we can *enforce it by design*, e.g., when we control data collection by synthesizing comparisons via separate "good" and "bad" generators. Nevertheless, our goal is not only to verify if the CPRD is representable with a BT model but to learn the CPRD. This motivates the next section where we analyze the learning objectives.

## 5. Learning CPRD

We now turn to a question of more immediate practical relevance: given i.i.d. triplets $(x, y^+, y^-)$ drawn from $P$, what objective should we optimize, and what does that objective actually recover and when it matches the ground-truth CPRD? We study two canonical approaches, generative modeling via MLE of the triplet distribution, and discriminative BT training via pairwise log-likelihood.

First, we will introduce the generative and discriminative learning objectives for learning the CPRD.

**The generative learning objective.** For a class of parametric distribution $P_\phi$ over triplets $(x, y^+, y^-)$, the generative learning objective is

$$\min_\phi \mathcal{L}_{\text{gen}}(\phi) = \mathbb{E}_P \left[ -\log P_\phi(x, y^+, y^-) \right]. \quad (10)$$

where $P_\phi(x, y^+, y^-)$ is the probability of the triplet $(x, y^+, y^-)$ under the distribution $P_\phi$. Here, we are trying to fit the full generative model for the triplet distribution $P$. Once we have $\hat{P}_\phi$, the learned preference distribution for any unordered pair $\{y, y'\}$ is given by

$$\hat{\omega}_\phi(y \succ y' \mid x) = \frac{\hat{P}_\phi(x, y, y')}{\hat{P}_\phi(x, y, y') + \hat{P}_\phi(x, y', y)}. \quad (11)$$

Since minimizing the generative learning objective is equivalent to minimizing the KL divergence between the true distribution $P$ and the generative model $P_\phi$ (White, 1982), the generative learning objective have a CPRD recovery guarantee; the learned preference distribution would converge to the true CPRD when the true CPRD belongs to the model space (realizable). On the other hand, the discriminative objective works directly at the preference level without explicitly modelling the triplet distribution $P$.

**The discriminative BT learning objective.** For a class of parametric score functions $r_\theta : \mathcal{X} \times \mathcal{Y} \to \mathbb{R}$, the discriminative BT learning objective is

$$\min_\theta \mathcal{L}_{\text{BT}}(\theta) = \mathbb{E}_P \left[ -\log P_{r_\theta}(y^+ \succ y^- \mid x) \right]. \quad (12)$$

where $P_{r_\theta}(y^+ \succ y^- \mid x) = \sigma(r_\theta(x, y^+) - r_\theta(x, y^-))$ is the probability of preferring $y^+$ over $y^-$ given $x$ under the BT model with score function $r_\theta$. Once we have $\hat{r}_\theta$, the learned preference distribution for any unordered pair $\{y, y'\}$ is given by

$$\hat{\omega}_\theta(y \succ y' \mid x) = \sigma(\hat{r}_\theta(x, y) - \hat{r}_\theta(x, y')). \quad (13)$$

In contrast with the generative objective, it is not straightforward to see how this objective would recover the true CPRD. In the next result, we show that this objective is also equivalent to minimizing the KL divergence between the true distribution and the preference distribution induced by the BT model but at the preference level instead of the triplet level.

**Definition 5.1** (Comparison distribution)**.** For any distribution $P$ over $\mathcal{X} \times \mathcal{Y} \times \mathcal{Y}$, the comparison distribution $\widetilde{P}$ is a distribution over $x$ and an unordered pair $\{y, y'\}$ with $y \neq y'$ with density

$$\widetilde{P}(x, \{y, y'\}) \propto P(x, y, y') + P(x, y', y). \quad (14)$$

This distribution essentially captures how often a pair of outcomes $y$ and $y'$ are compared together given the context $x$ in the data, regardless of the outcome.

**Theorem 5.2** (Decomposition of the discriminative BT learning objective). *The discriminative BT learning objective* (12) *can be decomposed as*

$$\mathcal{L}_{BT}(\theta) = C + Z \underset{(x,\{y,y'\})\sim\widetilde{P}}{\mathbb{E}} \Big[ D_{\mathrm{KL}}\big( \mathrm{Bern}(\omega_P(y \succ y' \mid x))$$

$$\| \mathrm{Bern}(P_{r_\theta}(y \succ y' \mid x)))\Big] \qquad (15)$$

*where $C$ is independent of $\theta$ and $\mathrm{Bern}(p)$ is the Bernoulli distribution with parameter $p$. $\widetilde{P}$ is a compairson distribution induced by $P$ and $Z$ is its normalizing constant. The CPRD $\omega_P$ in equation* (15) *is well-defined whenever $\widetilde{P}(x,\{y,y'\}) > 0$.*

*Proof in Appendix B.*

The KL divergence term in equation (15) is independent of the ordering of $y$ and $y'$ and therefore it is well-defined for any $(x, \{y, y'\})$. As a direct consequence of this Theorem, whenever the CPRD is BT model, the minimizer of the discriminative BT learning objective (12) would recover the true CPRD almost surely.

**Corollary 5.3.** *For any distribution $P$ such that its CPRD is representable by a BT model, the global minimizer $\hat{r}_\theta$ of the discriminative BT learning objective* (12) *satisfies*

$$P_{\hat{r}_\theta}(y \succ y' \mid x) = \omega_P(y \succ y' \mid x) \qquad (16)$$

*for $\widetilde{P}$-almost every unordered comparison $(x, \{y, y'\})$.*

*Proof in Appendix B.*

Combining with the previous result (Theorem 4.4), we have a recovery guarantee under the conditional independence assumption.

**Corollary 5.4** (Consistency under conditional independence). *Assume that $P$ satisfies positive–negative conditional independence. If the BT model family contains a parameter $\theta^*$ such that $P_{r_\theta^*}(y \succ y' \mid x) = \omega_P(y \succ y' \mid x)$, then*

1. *$\theta^*$ minimizes the discriminative BT learning objective* (12)

2. *Any global minimizer $\hat{\theta}$ of the discriminative BT learning objective* (12) *satisfies*

$$P_{\hat{r}_\theta}(y \succ y' \mid x) = \omega_P(y \succ y' \mid x) \qquad (17)$$

*for $\widetilde{P}$-almost every unordered comparison $(x, \{y, y'\})$.*

*Proof in Appendix B.*

To summarize, the generative and discriminative learning objectives both have a clean statistical interpretation of minimizing the KL divergence between the true distribution and the preference distribution induced by our model. The former is on the triplet level, while the latter is on the preference level. Whenever the model space is realizable, both objectives would recover the true CPRD almost surely. When the assumption fails, they converge to an explicit projection of the true CPRD on to the model family. For discriminative objective, this is a positive–negative conditional independent distribution.

## 6. Learning Score Functions from Pairwise Comparisons

We have established that the BT objective is a consistent learning objective for CPRD. However, in practice, we only have access to a finite number of triplets, it is not clear how fast the learned score function converges to the target score function (sample complexity).

In this section, we fix a score function $r^* : \mathcal{X} \times \mathcal{Y} \to \mathbb{R}$ as a target and study the impact of the data distribution $P$ on the sample complexity for recovering $r^*$. The first natural question is how to design a distribution of triplets to learn a desired score function. Recall from Theorem 4.4 that if a distribution $P$ satisfies positive–negative conditional independence, then its CPRD is given by a BT model with a score function $r(x, y) = \log \frac{p_+(y|x)}{p_-(y|x)}$. We can reverse engineer this: for any target score function $r^*(x, y)$, any positive–negative conditional independence distribution $P$ that satisfies $p_+(y \mid x) = \exp(r^*(x,y))p_-(y \mid x)$ would recover $r^*$ with the BT model. We call this pair of conditional distributions *BT-consistent* (with respect to $r^*$).

**Definition 6.1.** Let $r^* : \mathcal{X} \times \mathcal{Y} \to \mathbb{R}$ be a target score function. A pair of conditional distributions $p_+(\cdot \mid x)$ and $p_-(\cdot \mid x)$ is *BT-consistent* with respect to $r^*$ if

$$p_+(y \mid x) = \exp(r^*(x,y))p_-(y \mid x) \qquad (18)$$

for all $x$ and $y$.

The score function from fitting a BT model on any pair $(p_+, p_-)$ that is BT-consistent would recover $r^*(x, y)$. This answers our first question on how to design a distribution of triplets to learn a desired score function. Since, $p_-$ is not necessarily unique, do different choices of $(p_+, p_-)$ lead to different sample complexity? We will address this question in the following subsections. We start by defining the accuracy of a score function. Here, we measure the accuracy in terms of the agreement of the ordering of scores.

**Definition 6.2** (Margin). For any score function $r$ and any context $x$ and responses $y, y'$, the pairwise margin of $r$ is given by

$$\Delta_r(x; y, y') := r(x, y) - r(x, y') \qquad (19)$$

**Definition 6.3** (Accuracy of a score function). Let $Q$ be a distribution over $(x, y)$, we define $Q_{\text{pair}}$ as a distribution over $(x, y, y')$ where $x \sim Q_x$ and $y, y' \sim^{iid} Q_y(y \mid x)$. The accuracy of a score function $r$ with respect to the target score $r^*$ and $Q$ is defined as

$$\text{Acc}_Q(r) = \Pr_{x,y,y' \sim Q_{\text{pair}}} [\text{sign}(\Delta_r) = \text{sign}(\Delta_{r^*})] \quad (20)$$

The margin function $\Delta_r$ takes $(x, y, y')$ as input but here we drop them for convenience. The distribution $Q$ controls the support and density of the responses where we want to evaluate the accuracy. We also call it a test distribution. For example, $Q$ could be a uniform distribution over the response space or could be skewed toward the responses that we care about. Our goal is to learn a score function that has high accuracy. Our first observation is that whenever the estimation error of the margin is smaller than the margin then the ordering remains the same.

**Proposition 6.4** (Accuracy Bound). *For any score $r$,*

$$\text{Acc}_Q(r) \geq \Pr_{Q_{\text{pair}}} (|(\Delta_r - \Delta_{r^*})| \leq |\Delta_{r^*}|) \quad (21)$$

This is a direct consequence of the following lemma.

**Lemma 6.5** (Order Preservation). *For any real numbers $a$ and $b$, if $|b - a| \leq |a|$, then $a$ and $b$ have the same sign.*

From this Proposition, there are two factors that influence the accuracy: the **true margin** $|\Delta_{r^*}|$ and the **estimation error** of the margin $|\Delta_r - \Delta_{r^*}|$. The margin is a function of the target score which is fixed apriori. On the other hand, the estimation error depends on how well we are able to learn the score function. We show that the connectivity of the data distribution controls this estimation error. We first define the connectivity degree.

**Definition 6.6** (Connectivity). For a distribution $P$ and a hypothesis class $\mathcal{H}$ of score functions and the test distribution $Q$, the connectivity degree of $P$ with respect to $\mathcal{H}$ and $Q$ is defined as

$$\lambda_{\text{conn}}(P, Q; \mathcal{H}) = \inf_{f, g \in \mathcal{H}} \frac{\mathbb{E}_{\widetilde{P}}[(\Delta f - \Delta g)^2]}{\widetilde{\text{Var}}_Q[(f - g)]} \quad (22)$$

when where $\widetilde{P}$ is the comparison distribution induced by $P$ defined in Theorem 5.2 and $\widetilde{\text{Var}}_Q(r)$ is the variance of $r$ with respect to $Q$ defined as

$$\widetilde{\text{Var}}_Q(r) = \mathbb{E}_{x \sim Q_x} \text{Var}_{y \sim Q_y(y|x)}[r(x, y)] \quad (23)$$

The proposed connectivity degree is a general quantity that is inherent to the distribution of the triplets and the hypothesis class. In particular, in the classical tabular BT that aims to recover the ranking of $m$ items and the hypothesis class

$\mathcal{H} = \{\theta : \theta \in \mathbb{R}^m\}$, $\lambda_{\text{conn}}$ is equivalent to the Fiedler value of the Laplacian matrix of the comparison graph which appears in the bound from (Shah et al., 2016). In the linear BT where $\mathcal{H} = \{r : r(x, y) = w^\top \phi(x, y), w \in \mathbb{R}^d\}$ when $\phi(x, y) \in \mathbb{R}^d$ is a feature encoder, $\lambda_{\text{conn}}$ is the smallest eigenvalue of a covariance matrix. This compliments the prior result for linear BT in (Zhu et al., 2023; Shen et al., 2025). We provide the full details of these results in the Appendix C. Another nice property of this connectivity degree is that it is defined and is computable to any hypothesis class. In the experiment section, we compute this for a class of two-layer neural networks.

Now, we are ready to present the result for the estimation error. We will present the result for the realizable setting, and refer to Appendix C for the proofs and agnostic setting.

**Theorem 6.7** (Estimation error bound, realizable). *Let $r^*$ be the target score function, $\mathcal{H}$ be a hypothesis class such that $r^* \in \mathcal{H}$ and $\mathcal{H}$ is bounded by some constant $B$. Let $P$ be a triplet distribution that is BT-consistent with respect to $r^*$ and let $S$ be a set of $n$ samples drawn from $P$. Let $\hat{r}$ be the empirical risk minimizer of the BT learning objective on $S$ then there exists a constant $c_B, M_B > 0$ such that with probability at least $1 - \delta$ over $S$,*

$$\mathbb{E}_{Q_{\text{pair}}}[(\Delta_r - \Delta_{r^*})^2]$$
$$\lesssim \frac{1}{\lambda_{\text{conn}}} \left( \frac{1}{c_B} (\hat{\mathfrak{R}}_S(\mathcal{H}_{\text{pair}}) + M_B \sqrt{\frac{\log(2/\delta)}{n}}) \right) \quad (24)$$

*when $\hat{\mathfrak{R}}_S$ is an empirical Rademacher complexity of the class $\mathcal{H}_{\text{pair}} := \{r(x, y) - r(x, y') : r \in \mathcal{H}\}$ and $\lambda_{\text{conn}}$ is the connectivity degree of $P$ with respect to $\mathcal{H}$ and $Q$. We use $X \lesssim Y$ as a shorthand for $X \leq cY$ for some constant $c$.*

The estimation error is upper bounded by the product of a complexity term and $\frac{1}{\lambda_{\text{conn}}}$. This suggest that when the distribution has a larger connectivity degree, it would have a lower estimation error. Finally, we combine with the previous result to provide the accuracy bound.

**Theorem 6.8** (Accuracy bound, realizable case). *Let $\hat{r}$ be the empirical risk minizer of the BT learning objective, then there exists a constant $D > 0$ such that with probability at least $1 - \delta$,*

$$\text{Acc}_Q(\hat{r}) \geq \sup_{k>0} \left[ \underbrace{\Pr_Q(|\Delta_{r^*}| \geq k)}_{\text{margin}} - \underbrace{D \frac{\text{Comp}(\mathcal{H}, \delta)}{k^2 \lambda_{\text{conn}}}}_{\text{connectivity}} \right] \quad (25)$$

*where the complexity term is defined as in right hand side of the Theorem 6.7.*

**Summary.** Our analysis reveals two fundamental factors that govern learning efficiency from pairwise comparisons:

1. **Pairwise Margin**: Larger margins between response scores make pairs easier to rank correctly, even with moderate estimation error.

2. **Comparison Graph Connectivity**: Higher connectivity reduces estimation error by ensuring information propagates well across all response pairs.

These insights provide practitioners with a framework for understanding when pairwise preference learning will succeed: design data collection processes that induce both large margins and well-connected comparison graphs.

# 7. Experiments

We conduct experiments on synthetic data to validate our theoretical findings. With synthetic data, we have access to the ground-truth score function $r^*$, enabling us to precisely measure the effects of margin and connectivity on learning efficiency.

## 7.1. Setup.

**Context and Response Space.** Let $\mathcal{X} = \mathcal{Y} = \{x_1, x_2, \ldots, x_m\}$ be a finite set of contexts and responses where $x_i \in \mathbb{R}^d$. We use the same space for contexts and responses to simplify the synthetic setting. One could interpret the score function as a similarity function between items. We set $m = 16$ and $d = 128$ for simplicity. To generate the dataset, we sample $x_i$ from $\mathcal{N}(0, I_d)$.

**Ground-truth Score Function.** The ground-truth score is defined via a two-layer neural network $f^* : \mathbb{R}^d \to \mathbb{R}^e$ with hidden dimension $h = 32$ and output embedding dimension $e = 8$, using ReLU activations:

$$r^*(x, y) = \text{cosine-similarity}(f^*(x), f^*(y)), \quad (26)$$

ensuring scores lie in $[-1, 1]$.

**Triplet Generation.** We generate $n$ triplets $S = \{(x_i, y_i^+, y_i^-)\}_{i=1}^n$ by sampling $x_i$ uniformly from $\mathcal{X}$ and drawing $y_i^+$ and $y_i^-$ from a negative distribution $p^-(\cdot \mid x_i)$ and a positive distribution $p^+(\cdot \mid x_i)$ that is BT-consistent with $r^*$, i.e., $\log(p^+(y \mid x)/p^-(y \mid x)) = r^*(x, y)$. By default, we use a uniform negative distribution $p^-(\cdot \mid x) = \frac{1}{m}$. The validation set is also generated in the same way.

**Training and Evaluation.** We train a two-layer neural network with the same architecture as $f^*$ to minimize the empirical BT loss on $S$ (equation (5)). We use Adam optimizer with learning rate selected from $\{10^{-4}, 10^{-3}, 10^{-2}\}$. We train for a fixed number of epochs 200 and select hyperparameters via a validation set of 2048 triplets drawn from the same distribution based on validation loss. We report accuracy with respect to the uniform distribution $Q$ over all pairs $(x_i, x_j)$ with standard error over 5 random seeds.

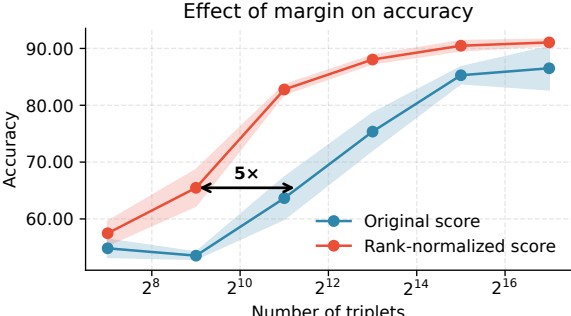

*Figure 1.* Effect of margin on learning efficiency. Learning from the rank-normalized score $r_{\text{rank}}^*$ (which has larger minimum margin) achieves higher accuracy than learning from $r^*$, especially when the number of triplets is small. The gap diminishes as sample size increases.

## 7.2. Larger Margin Leads to Higher Accuracy

Our theory suggests that larger score margins lead to higher accuracy, since moderate estimation errors are less likely to flip the ordering of well-separated pairs. To isolate the effect of margin from other factors, we compare learning from the original score $r^*$ against a *rank-normalized* version $r_{\text{rank}}^*$ that preserves the ordering but maximizes the minimum margin between any two responses.

**Rank normalization.** For each context $x$, we sort responses by $r^*(x, y)$ and assign:

$$r_{\text{rank}}^*(x, y) = -1 + \frac{2 \cdot \text{rank}(y)}{m}, \quad (27)$$

where $\text{rank}(y) \in \{1, \ldots, m\}$ is the rank of $y$ among all responses for context $x$ (ascending). This transformation preserves the ordering while maximizing the minimum pairwise margin to $2/m$. We construct BT-consistent distributions for both $r^*$ and $r_{\text{rank}}^*$ using the same uniform $p^-$.

**Results.** As shown in Figure 1, learning from $r_{\text{rank}}^*$ achieves substantially higher accuracy than learning from $r^*$, particularly in the small-sample regime. As $n$ increases, the gap narrows, consistent with our theory: with sufficient data, estimation error becomes small enough that even small margins suffice for correct ordering. To further validate our theory, we analyze accuracy on pairs with the smallest margins (bottom 10% and 30%). The accuracy gap between $r_{\text{rank}}^*$ and $r^*$ is most pronounced on these pairs, confirming that margin amplification primarily benefits pairs that are otherwise difficult to distinguish. See Appendix D for more details.

## 7.3. Low Connectivity Predicts Poor Accuracy

Our theory suggests that higher connectivity leads to lower estimation error. In this experiment, we fix the score func-

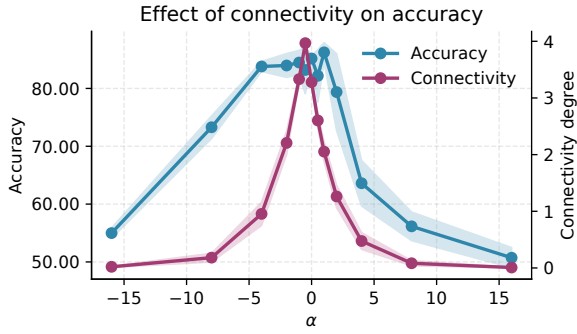

*Figure 2.* Effect of comparison graph connectivity on learning. High connectivity degress values are associated with high accuracy, though the relationship is not strictly monotonic.

tion $r^*$ and vary the $p^-$ to change the connectivity of the triplet distribution.

**Varying the negative distribution.** We fix $r^*$ and parameterize the negative distribution as $p^-(y \mid x) \propto \exp(\alpha \cdot r^*(x, y))$ for $\alpha \in [-16, 16]$. Different values of $\alpha$ induce different sampling strategies: i) $\alpha < 0$: Easy negatives, prefer sample with low scores ii) $\alpha = 0$: Uniform sampling, iii) $\alpha > 0$: Hard negatives, prefer sample with high scores. For each $\alpha$, we compute the corresponding BT-consistent $p^+$ and measure both accuracy and the connectivity degree $\lambda_{\text{conn}}(P^\alpha)$.

**Results.** As shown in Figure 2, extreme values of $\alpha$ correspond to overly easy or hard negatives which lead to low connectivity. Moreover, low connectivity reliably predicts poor accuracy, though the converse does not hold: high connectivity does not guarantee high accuracy. This asymmetry reflects the worst-case nature of our bound: low connectivity means some functions in the hypothesis class are hard to learn, but the actual target may not be among them.

### 7.4. Optimizing Connectivity Helps When Connectivity is the Bottleneck

We investigate whether directly optimizing $p^-$ to increase the connectivity degree improves learning.

**Method.** We scale the target reward by a factor of $\beta$, $r^*_\beta = \beta \cdot r^*(x, y)$ for different values of $\beta$. This provides a way to control the difficulty of the task where smaller $\beta$ corresponds to a task with small margin and large connectivity while a larger $\beta$ corresponds to a task with large margin and small connectivity. We compare two negative distributions

- **Baseline**: uniform negative distribution $p^-(\cdot \mid x) = \frac{1}{m}$

- **Optimized**: we optimize $p^-$ to maximize the connectivity degree $\lambda_{\text{conn}}(P^\beta)$.

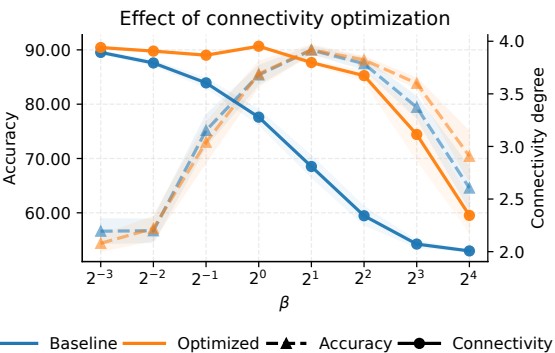

*Figure 3.* Effect of connectivity optimization on learning. Optimizing $p^-$ to increase the connectivity can help with the accuracy when the task is hard where $\beta$ is large.

**Results.** First, optimizing $p^-$ leads to higher connectivity than uniform sampling across all values of $\beta$. However, higher connectivity only improves accuracy in the *connectivity-dominated regime* (large $\beta$), where the margin is sufficient but connectivity is the bottleneck. In the *margin-dominated regime* (small $\beta$), the margin is too small for learning to succeed—no amount of connectivity optimization can compensate. This validates that connectivity can be a meaningful quantity to optimize in practice when designing a data distribution.

### 7.5. BT Converges to the KL Projection

Our theory predicts that when the true CPRD is non-BT, BT training does not recover the true CPRD itself, but rather its KL projection onto the BT family. To verify this mechanism, we construct a synthetic non-BT CPRD by perturbing a ground-truth BT CPRD in a way that breaks BT representability while preserving the same BT projection target $r^*$ (Appendix D.4). We then train BT models on samples from both the original and perturbed distributions.

*Table 1.* BT training under well-specified and misspecified CPRDs. Both settings converge toward the same BT projection target $r^*$, while the pairwise-probability error remains larger under the non-BT CPRD, reflecting irreducible model misspecification.

| $n$ | RMSE to $r^*$ ($\times 10^{-1}$) | Pairwise MAE ($\times 10^{-2}$) |
|---|---|---|
| **BT CPRD** | | |
| 128 | $2.422 \pm 0.302$ | $6.909 \pm 0.843$ |
| 512 | $2.450 \pm 0.267$ | $6.917 \pm 0.761$ |
| 2048 | $1.909 \pm 0.101$ | $5.354 \pm 0.265$ |
| 8192 | $1.520 \pm 0.118$ | $4.277 \pm 0.340$ |
| 32768 | $1.019 \pm 0.123$ | $2.879 \pm 0.344$ |
| 131072 | $0.983 \pm 0.138$ | $2.754 \pm 0.412$ |
| **Non-BT CPRD** | | |
| 128 | $2.534 \pm 0.300$ | $10.234 \pm 0.402$ |
| 512 | $2.446 \pm 0.301$ | $10.019 \pm 0.452$ |
| 2048 | $2.098 \pm 0.154$ | $9.211 \pm 0.179$ |
| 8192 | $1.500 \pm 0.091$ | $8.082 \pm 0.203$ |
| 32768 | $1.033 \pm 0.131$ | $7.202 \pm 0.433$ |
| 131072 | $0.727 \pm 0.128$ | $6.618 \pm 0.314$ |

**Results.** As the sample size grows, the learned reward converges to the same BT target $r^*$ in both settings, consistent with the KL projection view. At the same time, the pairwise-probability MAE remains larger in the non-BT setting, reflecting the irreducible misspecification.

### 7.6. Connectivity on Real Preference Datasets

Our theory predicts that good reward-model generalization requires both large reward margins and connectivity to the evaluation distribution. We test whether this trade-off appears on real preference datasets, and whether connectivity can serve as a practical diagnostic rather than only a theoretical quantity.

We train a linear reward head on frozen embeddings from `Llama-3.1-8B-Instruct` (Grattafiori et al., 2024) with a BT objective using 10K preference pairs from each training dataset: `HH-RLHF` (Bai et al., 2022a), `PKU-SafeRLHF` (Ji et al., 2025), `SHP` (Ethayarajh et al., 2022), and `UltraFeedback` (Cui et al.). All models are evaluated on the same `UltraFeedback` test set, re-labeled using `Skywork-Reward-V2-Llama-3.1-8B` (Liu et al., 2025a). This controlled setup allows us to measure both the learned reward margins and a test-distribution variant of the connectivity degree. Table 2 reports averages over five random seeds, with standard errors. Since the test distribution is not uniform over a test distribution $Q$ anymore, we discussed how we calculate the connectivity degree for this setting in Appendix D.3.

*Table 2.* Margin, connectivity, and accuracy for reward models trained on different preference datasets and evaluated on the Ultra-Feedback test distribution. Connectivity is reported at scale $10^{-4}$.

| Dataset | Margin | Conn. ($\times 10^{-4}$) | Acc. |
|---|---|---|---|
| HH-RLHF | $10.766 \pm 0.036$ | $3.30 \pm 0.05$ | $74.020 \pm 0.339$ |
| PKU-SafeRLHF | $10.527 \pm 0.017$ | $0.85 \pm 0.05$ | $74.790 \pm 0.545$ |
| SHP | $8.833 \pm 0.026$ | $15.14 \pm 0.29$ | $78.380 \pm 0.244$ |
| UltraFeedback | $11.255 \pm 0.029$ | $8.96 \pm 0.16$ | $82.170 \pm 0.129$ |

**Results.** The results show that connectivity helps explain reward-model performance beyond what is captured by margins alone. HH-RLHF and PKU-SafeRLHF achieve reasonably large margins, but have substantially lower connectivity to the UltraFeedback test distribution and underperform SHP. This is consistent with domain mismatch: HH-RLHF and PKU-SafeRLHF emphasize safety-related comparisons, whereas UltraFeedback contains broader general-purpose instruction-following comparisons. At the same time, connectivity alone is not sufficient. SHP has the highest connectivity, but UltraFeedback achieves the best accuracy because it also induces substantially larger margins. Overall, these results support our theory: strong generalization requires both adequate coverage of the relevant test-distribution com-

parisons and sufficiently large reward margins. They also show that connectivity is not merely a theoretical quantity; it can be computed on real preference datasets and helps explain accuracy differences that margin alone misses.

## 8. Discussion

We provide a framework for understanding preference learning from pairwise comparisons through a data-centric lens. The key object is the CPRD, which captures the preference information encoded in the triplet data distribution. We establish precise conditions under which the CPRD is representable by a BT model and analyze the learning efficiency of the BT objective. Our theoretical and empirical results identify margin and connectivity as the governing factors of learning efficiency.

These results suggest two diagnostics for preference data. First, among data collection strategies that are otherwise reasonable for the target task, one should prefer triplet distributions with better connectivity relative to the intended evaluation distribution, since such distributions provide better coverage of the pairwise variations that matter at test time. Second, our characterization of BT representability gives a concrete way to reason about model misspecification. In particular, BT is appropriate when the chosen and rejected responses satisfy the positive–negative conditional-independence structure. Strong violations of this structure, for example when a positive response is produced by editing a negative response, suggest that the BT model may be misspecified for the resulting preference data. Thus, our framework does not only describe when BT learning is statistically efficient, but also provides a diagnostic lens for deciding when BT is a reasonable modeling assumption.

Several practical challenges remain. Since the target score function is implicit, it is unclear how to evaluate or increase the margin directly in practice. Although connectivity is computable from data, developing actionable strategies for improving it remains an open question. Similarly, while our characterization reduces BT misspecification to a positive–negative conditional-independence question, testing this condition in high-dimensional preference data may be difficult; in simpler parametric or nonparametric settings, one could apply standard conditional-independence tests, such as Fisher's $z$-test or kernel-based tests. Finally, our analysis assumes that the target score function is known when designing the distribution, whereas in practice it is exactly what we want to learn. One possible relaxation is to allow distribution changes that also shift the target, provided the new target remains desirable. Developing practical data curation strategies that yield distributions which are both easy to learn from and induce good target score functions is a promising direction for future work.

## Acknowledgements

This work was supported in part by Bloomberg Data Science Fellowship, AFRL and DARPA via FA8750-23-2-1015, ONR via N00014-23-1-2368, and NSF via IIS-1955532 and IIS-1901403. The authors thank Korinna Fragkia and Andreas Kalavas for their insightful feedback on the draft.

## Impact Statement

This paper presents work whose goal is to advance the field of Machine Learning. There are many potential societal consequences of our work, none which we feel must be specifically highlighted here.

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

## A. Proofs for CPRD Representability

**Proposition 4.2** (CPRD BT factorization). *For a distribution $P$ over $\mathcal{X} \times \mathcal{Y} \times \mathcal{Y}$, the CPRD $\omega_P$ is representable by a BT model if and only if there exists a strictly positive function $h : \mathcal{X} \times \mathcal{Y} \to \mathbb{R}$ such that for all $x$ and all $y \neq y'$ with $P(x, y, y') + P(x, y', y) > 0$,*

$$\frac{P(x, y, y')}{P(x, y', y)} = \frac{h(x, y)}{h(x, y')}. \tag{7}$$

*Proof.* Fix $x$ and $y \neq y'$ with $P(x, y, y')P(x, y', y) > 0$. By the CPRD definition,

$$\frac{\omega_P(y \succ y' \mid x)}{\omega_P(y' \succ y \mid x)} = \frac{P(x, y, y')}{P(x, y', y)}. \tag{28}$$

($\Rightarrow$) If $\omega_P$ is representable by a BT model, then there exists a score function $r : \mathcal{X} \times \mathcal{Y} \to \mathbb{R}$ such that

$$\omega_P(y \succ y' \mid x) = \sigma\big(r(x, y) - r(x, y')\big). \tag{29}$$

In addition,

$$\omega_P(y' \succ y \mid x) = \sigma\big(r(x, y') - r(x, y)\big). \tag{30}$$

Write $\Delta = r(x, y) - r(x, y')$, by substituting this into equation (28) gives

$$\frac{P(x, y, y')}{P(x, y', y)} = \frac{\sigma(\Delta)}{\sigma(-\Delta)} = \exp(\Delta). \tag{31}$$

Setting $h(x, y) := \exp(r(x, y)) > 0$ yields (7).

($\Leftarrow$) Conversely, assume (7) holds for some $h > 0$ and define $r(x, y) := \log h(x, y)$. Then (7) becomes

$$\frac{P(x, y, y')}{P(x, y', y)} = \exp\big(r(x, y) - r(x, y')\big). \tag{32}$$

By the identity $\sigma(t)/\sigma(-t) = \exp(t)$, and using (28) we have

$$\frac{\omega_P(y \succ y' \mid x)}{\omega_P(y' \succ y \mid x)} = \frac{P(x, y, y')}{P(x, y', y)} = \frac{\sigma(\Delta)}{\sigma(-\Delta)} \tag{33}$$

Since $\omega_P(y \succ y' \mid x) + \omega_P(y' \succ y \mid x) = 1$ and $\sigma(\Delta) + \sigma(-\Delta) = 1$, we can conclude that $\omega_P(y \succ y' \mid x) = \sigma(\Delta)$ as desired. $\square$

**Theorem 4.4** (CPRD of conditionally independent distribution and BT model). *Let $P$ be a distribution over $\mathcal{X} \times \mathcal{Y} \times \mathcal{Y}$, then the following holds;*

1. *If $P$ satisfies positive–negative conditional independence, then the CPRD $\omega_P$ is representable by a BT model.*

2. *If the CPRD $\omega_P$ is representable by a BT model, then there exists a positive–negative conditional independence distribution $Q$ such that $\omega_Q = \omega_P$.*

*In addition, the suitable score function $r$ for a positive–negative conditional independence distribution $P$ is given by*

$$r(x, y) = \log \frac{p_+(y \mid x)}{p_-(y \mid x)}. \tag{9}$$

*Proof.* ($\Rightarrow$) Assume that $P$ is a positive–negative conditional independence distribution, then for any $x$ and $y \neq y'$, we can write $P$ as

$$P(x, y^+, y^-) = P_X(x)\, p_+(y^+ \mid x)\, p_-(y^- \mid x). \tag{34}$$

Therefore,

$$\frac{P(x, y, y')}{P(x, y', y)} = \frac{P_X(x)\, p_+(y \mid x)\, p_-(y' \mid x)}{P_X(x)\, p_+(y' \mid x)\, p_-(y \mid x)} \tag{35}$$

$$= \frac{p_+(y \mid x)\, p_-(y' \mid x)}{p_+(y' \mid x)\, p_-(y \mid x)} \tag{36}$$

$$= \frac{h(x, y)}{h(x, y')}. \tag{37}$$

where we define $h(x, y) := \frac{p_+(y|x)}{p_-(y|x)}$. Since we can write the ratio of the probability in this form, by Proposition 4.2, we can conclude that $\omega_P$ is representable by a BT model. In addition, recall from the proof of Proposition 4.2 that the score function $r$ is given by $r(x, y) = \log h(x, y)$. Therefore, we also have

$$r(x, y) = \log \frac{p_+(y \mid x)}{p_-(y \mid x)}. \tag{38}$$

($\Leftarrow$) Conversely, assume that the CPRD $\omega_P$ is representable by a BT model, then there exists a score function $r : \mathcal{X} \times \mathcal{Y} \to \mathbb{R}$ such that

$$\omega_P(y \succ y' \mid x) = \sigma(r(x, y) - r(x, y')). \tag{39}$$

We will construct a positive–negative conditional independence distribution $Q$ such that $\omega_Q = \omega_P$. For any $x$, assume that we have a base marginal distribution $\mu(\cdot \mid x)$ on $\mathcal{Y}$ with a full support. Define $q_-(y \mid x) = \mu(y \mid x)/Z_-(x)$ where $Z_-(x) = \int \mu(y \mid x) dy$ is value that ensure that $q_-(y \mid x)$ is a valid probability distribution. Define $q_+(y \mid x) = \mu(y \mid x) \exp(r(x, y))/Z_+(x)$ where $Z_+(x) = \int \mu(y \mid x) \exp(r(x, y)) dy$ is value that ensure that $q_+(y \mid x)$ is a valid probability distribution. The distribution $Q$ is given by

$$Q(x, y^+, y^-) = P_X(x)\, q_+(y^+ \mid x)\, q_-(y^- \mid x). \tag{40}$$

We can see that

$$\frac{Q(x, y, y')}{Q(x, y', y)} = \frac{q_+(y \mid x)\, q_-(y' \mid x)}{q_+(y' \mid x)\, q_-(y \mid x)} = \frac{h'(x, y)}{h'(x, y')} \tag{41}$$

where we define $h'(x, y) := \frac{q_+(y|x)}{q_-(y|x)}$. Since we can write the ratio of the probability in this form, by Proposition 4.2, we can conclude that $\omega_Q$ is representable by a BT model. Similarly, from the proof of Proposition 4.2, the score function $r_Q$ of the BT model from fitting with the distribution $Q$ is given by $r_Q(x, y) = \log h'(x, y)$,

$$r_Q(x, y) = \log \frac{q_+(y \mid x)}{q_-(y \mid x)} \tag{42}$$

$$= \log \frac{\mu(y \mid x) \exp(r(x, y))/Z_+(x)}{\mu(y \mid x)/Z_-(x)} \tag{43}$$

$$= r(x, y) + \log \frac{Z_-(x)}{Z_+(x)}. \tag{44}$$

Finally, we have

$$\omega_Q(y \succ y' \mid x) = \sigma(r_Q(x, y) - r_Q(x, y')) \tag{45}$$

$$= \sigma(r(x, y) + \log \frac{Z_-(x)}{Z_+(x)}) - (r(x, y') + \log \frac{Z_-(x)}{Z_+(x)})) \tag{46}$$

$$= \sigma(r(x, y) - r(x, y')) \tag{47}$$

$$= \omega_P(y \succ y' \mid x). \tag{48}$$

Therefore, we have $\omega_Q = \omega_P$ as desired. $\qquad\square$

## B. Proofs for Learning CPRD

**Theorem 5.2** (Decomposition of the discriminative BT learning objective). *The discriminative BT learning objective* (12) *can be decomposed as*

$$\mathcal{L}_{BT}(\theta) = C + Z \underset{(x,\{y,y'\}) \sim \widetilde{P}}{\mathbb{E}} \Big[ D_{\mathrm{KL}}\big( \mathrm{Bern}(\omega_P(y \succ y' \mid x))$$

$$\| \, \mathrm{Bern}(P_{r_\theta}(y \succ y' \mid x))) \Big] \tag{15}$$

*where $C$ is independent of $\theta$ and $\mathrm{Bern}(p)$ is the Bernoulli distribution with parameter $p$. $\widetilde{P}$ is a compairson distribution induced by $P$ and $Z$ is its normalizing constant. The CPRD $\omega_P$ in equation* (15) *is well-defined whenever $\widetilde{P}(x, \{y, y'\}) > 0$.*

*Proof.* We will start by writing the proof for the finite case when $\mathcal{X}$ and $\mathcal{Y}$ are finite. The continuous case follows by replacing sums with integrals.

The main idea is to write the objective in terms of unordered comparisons. For convenience, let $\mathcal{I} \subset \mathcal{X} \times \mathcal{Y} \times \mathcal{Y}$ be any index set such that for every $x \in \mathcal{X}$ and every distinct $y \neq y' \in \mathcal{Y}$, exactly one of $(x, y, y')$ and $(x, y', y)$ belongs to $\mathcal{I}$. Recall the definition of a comparison distribution $\widetilde{P}$,

$$\widetilde{P}(x, \{y, y'\}) = \frac{P(x, y, y') + P(x, y', y)}{Z}, \tag{49}$$

where $Z = \sum_{(x,y,y') \in \mathcal{I}} P(x, y, y') + P(x, y', y)$ is a normalizing constant. This distribution has a support over the this set $\mathcal{I}$. With this distribution, we can group the sum over $(y, y')$ and $(y', y)$ in the BT objectiveto obtain

$$\mathcal{L}_{BT}(\theta) = - \sum_{x \in \mathcal{X}, y, y' \in \mathcal{Y}} P(x, y, y') \log P_{r_\theta}(y \succ y' \mid x) \tag{50}$$

$$= - \sum_{(x,y,y') \in \mathcal{I}} P(x, y, y') \log P_{r_\theta}(y \succ y' \mid x) - \sum_{x \in \mathcal{X}, y \in \mathcal{Y}} P(x, y, y) \log P_{r_\theta}(y \succ y \mid x) \tag{51}$$

Since $P_{r_\theta}(y \succ y \mid x) = \sigma(0) = 0.5$, the second term is a constant that only depends on the distribution $P$ and we can write

$$\mathcal{L}_{BT}(\theta) = C - \sum_{(x,y,y') \in \mathcal{I}} P(x, y, y') \log P_{r_\theta}(y \succ y' \mid x) \tag{52}$$

$$= C - \sum_{(x,y,y') \in \mathcal{I}} P(x, y, y')(\log P_{r_\theta}(y \succ y' \mid x) + P(x, y', y) \log(P_{r_\theta}(y' \succ y \mid x))) \tag{53}$$

$$= C - Z \sum_{(x,y,y') \in \mathcal{I}} \widetilde{P}(x, \{y, y'\}) \Big( \frac{P(x, y, y')}{Z\widetilde{P}(x, \{y, y'\})} \log P_{r_\theta}(y \succ y' \mid x) \tag{54}$$

$$+ \frac{P(x, y', y)}{Z\widetilde{P}(x, \{y, y'\})} \log P_{r_\theta}(y' \succ y \mid x) \Big) \tag{55}$$

Recall the definition of the CPRD,

$$\omega_P(y \succ y' \mid x) = \frac{P(x, y, y')}{P(x, y', y) + P(x, y, y')} = \frac{P(x, y, y')}{Z\widetilde{P}(x, \{y, y'\})}. \tag{56}$$

In addition, we note that a cross entropy between two Bernoulli distributions $\mathrm{Bern}(p)$ and $\mathrm{Bern}(q)$ is given by

$$\mathrm{CE}(\mathrm{Bern}(p), \mathrm{Bern}(q)) = -p \log q - (1 - p) \log(1 - q) \tag{57}$$

which is exactly the term in the (55). To see that it is a Bernoulli distribution, we use the fact that $P_{r_\theta}(y \succ y' \mid x) + P_{r_\theta}(y' \succ y \mid x) = 1$ and $\omega_P(y \succ y' \mid x) + \omega_P(y' \succ y \mid x) = 1$. Therefore, we can write

$$-\frac{P(x,y,y')}{Z\widetilde{P}(x,\{y,y'\})}\log P_{r_\theta}(y \succ y' \mid x) - \frac{P(x,y',y)}{Z\widetilde{P}(x,\{y,y'\})}\log P_{r_\theta}(y' \succ y \mid x) \tag{58}$$

$$= \mathrm{CE}(\mathrm{Bern}(\omega_P(y \succ y' \mid x)), \mathrm{Bern}(P_{r_\theta}(y \succ y' \mid x))). \tag{59}$$

Substitute back, we have

$$\mathcal{L}_{\mathrm{BT}}(\theta) = C + Z \sum_{(x,y,y')\in\mathcal{I}} \widetilde{P}(x,\{y,y'\}) \mathrm{CE}(\mathrm{Bern}(\omega_P(y \succ y' \mid x)), \mathrm{Bern}(P_{r_\theta}(y \succ y' \mid x))). \tag{60}$$

Since cross entropy decomposes as

$$\mathrm{CE}(p,q) = H(p) + D_{\mathrm{KL}}(p\|q), \tag{61}$$

where $H(p)$ is the entropy and $D_{\mathrm{KL}}(p\|q)$ is the KL divergence, we can write the objective as

$$\mathcal{L}_{\mathrm{BT}}(\theta) = C + Z \sum_{(x,y,y')\in\mathcal{I}} \widetilde{P}(x,\{y,y'\}) \Big( H(\mathrm{Bern}(\omega_P(y \succ y' \mid x))) \tag{62}$$

$$+ D_{\mathrm{KL}}(\mathrm{Bern}(\omega_P(y \succ y' \mid x))\| \mathrm{Bern}(P_{r_\theta}(y \succ y' \mid x)))\Big). \tag{63}$$

The entropy term is independent of $\theta$ and is a constant term that depends on the true CPRD $\omega_P$. Finally, we can write the objective as

$$\mathcal{L}_{\mathrm{BT}}(\theta) = C + Z\mathbb{E}_{(x,\{y,y'\})\sim\widetilde{P}}[D_{\mathrm{KL}}(\mathrm{Bern}(\omega_P(y \succ y' \mid x))\| \mathrm{Bern}(P_{r_\theta}(y \succ y' \mid x)))]. \tag{64}$$

$\square$

**Corollary 5.3.** *For any distribution $P$ such that its CPRD is representable by a BT model, the global minimizer $\hat{r}_\theta$ of the discriminative BT learning objective* (12) *satisfies*

$$P_{\hat{r}_\theta}(y \succ y' \mid x) = \omega_P(y \succ y' \mid x) \tag{16}$$

*for $\widetilde{P}$-almost every unordered comparison $(x,\{y,y'\})$.*

*Proof.* From Theorem 5.2, we have

$$\mathcal{L}_{\mathrm{BT}}(\theta) = C + Z\mathbb{E}_{(x,\{y,y'\})\sim\widetilde{P}}[D_{\mathrm{KL}}(\mathrm{Bern}(\omega_P(y \succ y' \mid x))\| \mathrm{Bern}(P_{r_\theta}(y \succ y' \mid x)))]. \tag{65}$$

By the property of the KL divergence, the term $D_{\mathrm{KL}}(\mathrm{Bern}(\omega_P(y \succ y' \mid x))\| \mathrm{Bern}(P_{r_\theta}(y \succ y' \mid x)))$ is always non-negative and vanishes if and only if its arguments coincide. Since $\omega_P$ is representable by a BT model, we know that there exists $\theta^*$ such that $P_{r_\theta^*}(y \succ y' \mid x) = \omega_P(y \succ y' \mid x)$ which make the KL term vanish. As a result, $\mathcal{L}_{\mathrm{BT}}(\theta^*) = C$ which is the global minimum due to the non-negativity of the KL divergence term.

For any other global minimizer $\hat{r}_\theta$, we can't have the objective value smaller than this global minimum, we must have $\mathcal{L}_{\mathrm{BT}}(\hat{r}_\theta) = C$ and therefore it must be the case that

$$P_{\hat{r}_\theta}(y \succ y' \mid x) = \omega_P(y \succ y' \mid x) \tag{66}$$

for $\widetilde{P}$-almost every unordered comparison $(x,\{y,y'\})$. $\square$

**Corollary 5.4** (Consistency under conditional independence)**.** *Assume that $P$ satisfies positive–negative conditional independence. If the BT model family contains a parameter $\theta^*$ such that $P_{r_\theta^*}(y \succ y' \mid x) = \omega_P(y \succ y' \mid x)$, then*

*1. $\theta^*$ minimizes the discriminative BT learning objective* (12)

2. *Any global minimizer $\hat{\theta}$ of the discriminative BT learning objective* (12) *satisfies*

$$P_{\hat{r}_\theta}(y \succ y' \mid x) = \omega_P(y \succ y' \mid x) \tag{17}$$

*for $\widetilde{P}$-almost every unordered comparison $(x, \{y, y'\})$.*

*Proof.* Recall from Theorem 4.4, if the distribution $P$ satisfies positive–negative conditional independence, then its CPRD is a representable by a BT model. Therefore, the global minimizer of the discriminative BT learning objective (12) would exactly recover the true CPRD almost surely. □

## C. Sample Complexity

**Theorem C.1** (Estimation error bound). *Let $r^*$ be a target score function, $\mathcal{H}$ be a hypothesis class which is bounded by some constant $B$. Let $P$ be a triplet distribution that is BT-consistent with respect to $r^*$ and let $S$ be a set of $n$ samples drawn from $P$. Let $r_{opt} \in \arg\min_{r \in \mathcal{H}} L_{BT}(r)$ is the optimal score function in $\mathcal{H}$ that minimize the population BT objective with respect to $P$. Let $\hat{r}$ be the empirical risk minimizer of the empirical BT learning objective on $S$ then there exists a constant $c_B, M_B > 0$ such that with probability at least $1 - \delta$ over $S$,*

$$\mathbb{E}_{Q_{pair}}[(\Delta_{\hat{r}} - \Delta_{r^*})^2] \lesssim \frac{1}{\lambda_{conn}}((\frac{\epsilon_{misspec}}{c_B})^2 + \frac{1}{c_B}(\hat{\mathfrak{R}}_S(\mathcal{H}_{pair}) + M_B\sqrt{\frac{\log(2/\delta)}{n}})) + \mathbb{E}_{Q_{pair}}[(\Delta_{r_{opt}} - \Delta_{r^*})^2] \tag{67}$$

*$\lambda_{conn}$ is the connectivity degree, $\hat{\mathfrak{R}}_S(\mathcal{H}_{pair})$ is the empirical Rademacher complexity of the class $\mathcal{H}_{pair} := \{r(x, y) - r(x, y') : r \in \mathcal{H}\}$ and $\epsilon_{misspec}^2 = \mathbb{E}_{\widetilde{P}}[P_{r^*}(y \succ y' \mid x) - P_{r_{opt}}(y \succ y' \mid x)^2]$ is the misspecification error.*

*Proof.* **Step 1: Transition from $\Delta_{\hat{r}} - \Delta_{r^*}$ to $\Delta_{\hat{r}} - \Delta_{r_{opt}}$.**
Our goal is to bound $\mathbb{E}_{Q_{pair}}[(\Delta_{\hat{r}} - \Delta_{r^*})^2]$. We know that the empirical risk minimizer $\hat{r}$ would generally converge to $r_{opt}$. Therefore, we first bound the difference $\Delta_{\hat{r}} - \Delta_{r^*}$ into the difference $\Delta_{\hat{r}} - \Delta_{r_{opt}}$ and the difference $\Delta_{r_{opt}} - \Delta_{r^*}$.

$$\Delta_{\hat{r}} - \Delta_{r^*} = (\Delta_{\hat{r}} - \Delta_{r_{opt}}) + (\Delta_{r_{opt}} - \Delta_{r^*}) \tag{68}$$

Taking the expectation over $Q_{pair}$, and applying the Minkowski inequality,

$$\mathbb{E}_{Q_{pair}}[(\Delta_{\hat{r}} - \Delta_{r^*})^2] \leq 2\mathbb{E}_{Q_{pair}}[(\Delta_{\hat{r}} - \Delta_{r_{opt}})^2] + 2\mathbb{E}_{Q_{pair}}[(\Delta_{r_{opt}} - \Delta_{r^*})^2]. \tag{69}$$

The second term $\mathbb{E}_{Q_{pair}}[(\Delta_{r_{opt}} - \Delta_{r^*})^2]$ is the difference between the margin of the optimal score function in $\mathcal{H}$ and the true margin. This is a constant term that depends on the hypothesis class and the true score function and is zero in the realizable case. The next step is to bound the first term in this equation.

**Step 2: Transition from the expectation over $Q_{pair}$ to the expectation over $\widetilde{P}$.** Our accuracy is evaluated on the distribution $Q_{pair}$ but the BT objective is evaluated on the distribution $\widetilde{P}$. This part of the proof aim to connect them. Here we use the definition of the connectivity function and the two-sample variance identity.

Recall that we want to bound

$$\mathbb{E}_{Q_{pair}}[(\Delta_{\hat{r}} - \Delta_{r_{opt}})^2] = \mathbb{E}_{x \sim Q_x} \mathbb{E}_{y,y' \sim Q_y(y|x)}[(\Delta_{\hat{r}}(x, y, y') - \Delta_{r_{opt}}(x, y, y'))^2] \tag{70}$$

$$= \mathbb{E}_{x \sim Q_x} \mathbb{E}_{y,y' \sim Q_y(y|x)}[(\hat{r}(x, y) - \hat{r}(x, y')) - (r_{opt}(x, y) - r_{opt}(x, y'))]^2 \tag{71}$$

$$= \mathbb{E}_{x \sim Q_x} \mathbb{E}_{y,y' \sim Q_y(y|x)}[(\hat{r} - r_{opt})(x, y) - (\hat{r} - r_{opt})(x, y')]^2 \tag{72}$$

$$= 2\mathbb{E}_{x \sim Q_x} \text{Var}_{y \sim Q_y(y|x)}[(\hat{r} - r_{opt})(x, y)] \tag{73}$$

$$= 2\widetilde{\text{Var}}_Q[(\hat{r} - r_{opt})] \tag{74}$$

The second to last equality follows from the two-sample variance identity and the final equality follows from the definition of $\widetilde{\text{Var}}_Q$.

By the definition of the connectivity,

$$\lambda_{conn} = \inf_{f,g \in \mathcal{H}} \frac{\mathbb{E}_{\widetilde{P}}[(\Delta f - \Delta g)^2]}{\widetilde{\text{Var}}_Q[(f - g)]} \leq \frac{\mathbb{E}_{\widetilde{P}}[(\Delta_{\hat{r}} - \Delta_{r_{opt}})^2]}{\widetilde{\text{Var}}_Q[(\hat{r} - r_{opt})]} \tag{75}$$

Therefore,

$$\mathbb{E}_{Q_{\text{pair}}}[(\Delta_{\hat{r}} - \Delta_{r_{\text{opt}}})^2] \leq 2\widetilde{\text{Var}}_Q[(\hat{r} - r_{\text{opt}})] \leq \frac{2}{\lambda_{\text{conn}}}\mathbb{E}_{\widetilde{P}}[(\Delta_{\hat{r}} - \Delta_{r_{\text{opt}}})^2] \tag{76}$$

**Step 3: Bound $\mathbb{E}_{\widetilde{P}}[(\Delta_{\hat{r}} - \Delta_{r_{\text{opt}}})^2]$ in terms of the polulation risk $\mathcal{L}_{\text{BT}}(\hat{r}) - \mathcal{L}_{\text{BT}}(r_{\text{opt}})$.**
To relate the difference between the margin $\Delta_{\hat{r}} - \Delta_{r_{\text{opt}}}$ to the difference in the population risk $\mathcal{L}_{\text{BT}}(\hat{r}) - \mathcal{L}_{\text{BT}}(r_{\text{opt}})$, we will use the property of the BT objective and show that the loss is strongly convex. Recall from proof of Theorem 5.2, the BT objective can be decomposed as

$$\mathcal{L}_{\text{BT}}(r) = C + Z\mathbb{E}_{(x,\{y,y'\})\sim\widetilde{P}}\left[\text{CE}(\text{Bern}(\omega_P(y \succ y' \mid x)), \text{Bern}(P_r(y \succ y' \mid x)))\right]. \tag{77}$$

For each $(x, \{y, y'\})$, we define $g_{x,y,y'} : \mathbb{R} \to \mathbb{R}$,

$$g_{x,y,y'}(a) = \text{CE}(\text{Bern}(\omega_P(y \succ y' \mid x)), \text{Bern}(\sigma(a))) \tag{78}$$
$$= -\omega_P \log \sigma(a) - (1 - \omega_P) \log(1 - \sigma(a)) \tag{79}$$

We write $\omega_P$ for $\omega_P(y \succ y' \mid x)$ for convenience. With this definition, we can write the BT objective as

$$\mathcal{L}_{\text{BT}}(r) = C + Z\mathbb{E}_{(x,\{y,y'\})\sim\widetilde{P}}\left[g_{x,y,y'}(\Delta_r(x, y, y'))\right]. \tag{80}$$

We will write $\Delta_r$ for $\Delta_r(x, y, y')$ for convenience from now. Next, we will show that $g_{x,y,y'}$ is strongly convex with respect to $a$. We can check this by taking the derivative. Using the property that $\sigma'(a) = \sigma(a)(1 - \sigma(a))$, we have the first derivative

$$g'_{x,y,y'}(a) = \sigma(a) - \omega_P \tag{81}$$

and the second derivative

$$g''_{x,y,y'}(a) = \sigma(a)(1 - \sigma(a)) \tag{82}$$

Since the score function $r$ is bounded by a constant $B$, for any $(x, \{y, y'\})$, we know that $|\Delta_r| = |r(x, y) - r(x, y')| \leq 2B$. Now, for $a$ that is bounded above by a constant $2B$, we know that

$$\sigma(a)(1 - \sigma(a)) \geq \sigma(2B)(1 - \sigma(2B)) := c_B > 0 \tag{83}$$

With this assumption, we can conclude that $g_{x,y,y'}$ is $c_B$ strongly convex. Recall that if $f$ is $c$ strongly convex then for any $x, y$, we have

$$f(x) - f(y) \geq f'(y)(x - y) + \frac{c}{2}(x - y)^2. \tag{84}$$

Apply this inequality on $g_{x,y,y'}$ with $\Delta_{\hat{r}}(x, y, y')$ and $\Delta_{r_{\text{opt}}}(x, y, y')$ and drop $(x, y, y')$ for convenience.

$$g(\Delta_{\hat{r}}) - g(\Delta_{r_{\text{opt}}}) \geq g'(\Delta_{r_{\text{opt}}})(\Delta_{\hat{r}} - \Delta_{r_{\text{opt}}}) + \frac{c_B}{2}(\Delta_{\hat{r}} - \Delta_{r_{\text{opt}}})^2. \tag{85}$$

Now, we take the expectation over the distribution $\widetilde{P}$, we have

$$\frac{1}{Z}(\mathcal{L}_{\text{BT}}(\hat{r}) - \mathcal{L}_{\text{BT}}(r_{\text{opt}})) \geq \mathbb{E}_{\widetilde{P}}[g'(\Delta_{r_{\text{opt}}})(\Delta_{\hat{r}} - \Delta_{r_{\text{opt}}})] + \frac{c_B}{2}\mathbb{E}_{\widetilde{P}}[(\Delta_{\hat{r}} - \Delta_{r_{\text{opt}}})^2] \tag{86}$$

Rearranging,

$$\frac{c_B}{2}\mathbb{E}_{\widetilde{P}}[(\Delta_{\hat{r}} - \Delta_{r_{\text{opt}}})^2] \leq \frac{1}{Z}(\mathcal{L}_{\text{BT}}(\hat{r}) - \mathcal{L}_{\text{BT}}(r_{\text{opt}})) - \mathbb{E}_{\widetilde{P}}[g'(\Delta_{r_{\text{opt}}})(\Delta_{\hat{r}} - \Delta_{r_{\text{opt}}})] \tag{87}$$

$$\leq \frac{1}{Z}(\mathcal{L}_{\text{BT}}(\hat{r}) - \mathcal{L}_{\text{BT}}(r_{\text{opt}})) + |\mathbb{E}_{\widetilde{P}}[g'(\Delta_{r_{\text{opt}}})(\Delta_{\hat{r}} - \Delta_{r_{\text{opt}}})]| \tag{88}$$

We can bound the second term by using the Cauchy-Schwarz inequality,

$$|\mathbb{E}_{\widetilde{P}}[g'(\Delta_{r_{\text{opt}}})(\Delta_{\hat{r}} - \Delta_{r_{\text{opt}}})]| \leq \sqrt{\mathbb{E}_{\widetilde{P}}[g'(\Delta_{r_{\text{opt}}})^2]}\sqrt{\mathbb{E}_{\widetilde{P}}[(\Delta_{\hat{r}} - \Delta_{r_{\text{opt}}})^2]} \tag{89}$$

From our derivation, we know that $g'(a) = \omega_P - \sigma(a)$, so

$$\mathbb{E}_{\widetilde{P}}[g'(\Delta_{r_{\text{opt}}})^2] = \mathbb{E}_{\widetilde{P}}[\omega_P - \sigma(\Delta_{r_{\text{opt}}})]^2 \tag{90}$$

Since $P$ is BT-consistent, we have

$$\omega_P(y \succ y' \mid x) = P_{r^*}(y \succ y' \mid x) = \sigma(\Delta_{r^*}(x, y, y')) \tag{91}$$

Therefore, $\mathbb{E}_{\widetilde{P}}[g'(\Delta_{r_{\text{opt}}})^2]$ is the same as a misspecification error $\epsilon_{\text{misspec}}^2$ of $r_{\text{opt}}$.

$$|\mathbb{E}_{\widetilde{P}}[g'(\Delta_{r_{\text{opt}}})(\Delta_{\hat{r}} - \Delta_{r_{\text{opt}}})]| \leq \epsilon_{\text{misspec}} \sqrt{\mathbb{E}_{\widetilde{P}}[(\Delta_{\hat{r}} - \Delta_{r_{\text{opt}}})^2]} \tag{92}$$

Let $A = \sqrt{\mathbb{E}_{\widetilde{P}}[(\Delta_{\hat{r}} - \Delta_{r_{\text{opt}}})^2]}$, our goal is to bound $A^2$. Substitute the result from the Cauchy-Schwarz inequality back, we have

$$\frac{c_B}{2} A^2 \leq \frac{1}{Z}(\mathcal{L}_{\text{BT}}(\hat{r}) - \mathcal{L}_{\text{BT}}(r_{\text{opt}})) + \epsilon_{\text{misspec}} A \tag{93}$$

Solving this quadratic inequality, we have that

$$A \leq \frac{\epsilon_{\text{misspec}} + \sqrt{\epsilon_{\text{misspec}}^2 + \frac{2c_B}{Z}(\mathcal{L}_{\text{BT}}(\hat{r}) - \mathcal{L}_{\text{BT}}(r_{\text{opt}}))}}{c_B} \tag{94}$$

With Minkowski inequality, we have that

$$A^2 \leq 2(\frac{\epsilon_{\text{misspec}}}{c_B})^2 + 2((\frac{\epsilon_{\text{misspec}}}{c_B})^2 + \frac{2}{Zc_B}(\mathcal{L}_{\text{BT}}(\hat{r}) - \mathcal{L}_{\text{BT}}(r_{\text{opt}}))) \tag{95}$$

Finally, we have

$$\mathbb{E}_{\widetilde{P}}[(\Delta_{\hat{r}} - \Delta_{r_{\text{opt}}})^2] \leq 4((\frac{\epsilon_{\text{misspec}}}{c_B})^2 + \frac{1}{Zc_B}(\mathcal{L}_{\text{BT}}(\hat{r}) - \mathcal{L}_{\text{BT}}(r_{\text{opt}}))) \tag{96}$$

**Step 4: Use the uniform convergence result to bound the difference in the popuation risk $\mathcal{L}_{\text{BT}}(\hat{r}) - \mathcal{L}_{\text{BT}}(r_{\text{opt}})$**

This is the final step to bound the estimation error. Let $\widehat{\mathcal{L}}_{\text{BT}}(r)$ be the empirical risk of the BT objective with $n$ samples drawn from $P$. We write the difference in the population risk as

$$\mathcal{L}_{\text{BT}}(\hat{r}) - \mathcal{L}_{\text{BT}}(r_{\text{opt}}) = (\mathcal{L}_{\text{BT}}(\hat{r}) - \widehat{\mathcal{L}}_{\text{BT}}(\hat{r})) + (\widehat{\mathcal{L}}_{\text{BT}}(\hat{r}) - \widehat{\mathcal{L}}_{\text{BT}}(r_{\text{opt}})) + (\widehat{\mathcal{L}}_{\text{BT}}(r_{\text{opt}}) - \mathcal{L}_{\text{BT}}(r_{\text{opt}})) \tag{97}$$

$$\leq (\mathcal{L}_{\text{BT}}(\hat{r}) - \widehat{\mathcal{L}}_{\text{BT}}(\hat{r})) + (\widehat{\mathcal{L}}_{\text{BT}}(r_{\text{opt}}) - \mathcal{L}_{\text{BT}}(r_{\text{opt}})) \tag{98}$$

$$\leq 2 \sup_{r \in \mathcal{H}} |\mathcal{L}_{\text{BT}}(r) - \widehat{\mathcal{L}}_{\text{BT}}(r)| \tag{99}$$

This holds since $\hat{r}$ is the empirical risk minimizer therefore $\widehat{\mathcal{L}}_{\text{BT}}(\hat{r}) \leq \widehat{\mathcal{L}}_{\text{BT}}(r_{\text{opt}})$. We will bound the right hand side term using the standard Rademacher complexity result. As a quick recap, an empirical Rademacher complexity of a class $\mathcal{F}$ on the sample $S = \{z_i\}_{i=1}^n$ is defined as

$$\hat{\mathfrak{R}}_S(\mathcal{F}) = \mathbb{E}_\sigma \left[\sup_{f \in \mathcal{F}} \frac{1}{n} \sum_{i=1}^n \sigma_i f(z_i)\right] \tag{100}$$

where $\sigma_i$ are i.i.d. Rademacher random variables. The uniform convergence result states that if the class of loss function $\mathcal{G} := \{z \mapsto \ell(f, z) : f \in \mathcal{H}\}$ is bounded by some constant $M$, then with probability at least $1 - \delta$ over the sample $S$, we have

$$\sup_{f \in \mathcal{F}} |\mathbb{E}[\ell(f, z)] - \frac{1}{n} \sum_{z_i \in S} \ell(f, z_i)| \leq 2\hat{\mathfrak{R}}_S(\mathcal{G}) + 3M\sqrt{\frac{\log(2/\delta)}{n}} \tag{101}$$

Let $\mathcal{H}_{\text{pair}} = \{r(x, y) - r(x, y') : r \in \mathcal{H}\}$ be the class of pairwise score differences of score functions in $\mathcal{H}$. In our case, the class of loss function is given by $\ell \circ \mathcal{H}_{\text{pair}} = \{(x, y, y') \mapsto g_{x,y,y'}(r(x, y) - r(x, y')) : r \in \mathcal{H}\}$ since the BT objective is decomposed as

$$\mathcal{L}_{\text{BT}}(r) = C + Z\mathbb{E}_{(x, \{y, y'\}) \sim \widetilde{P}}[g_{x,y,y'}(r(x, y) - r(x, y'))] \tag{102}$$

Next, we will show that the loss is bounded by some constant $M$. From the definition

$$g_{x,y,y'}(a) = -\omega_P \log \sigma(a) - (1 - \omega_P) \log(1 - \sigma(a)) \tag{103}$$

$$\leq -\log(\min\{\sigma(a), 1 - \sigma(a)\}) \tag{104}$$

$$\leq \log(1 + \exp(2B)) := M_B \tag{105}$$

The final step is from our assumption that $r(x, y) \leq B$ for all $x, y$ so that $|r(x, y) - r(x, y')| \leq 2B$.

Therefore, from the uniform convergence from Rademacher complexity, with probability at least $1 - \delta$, we have

$$\sup_{r \in \mathcal{H}} \frac{1}{|Z|} |\mathcal{L}_{\text{BT}}(r) - \widehat{\mathcal{L}}_{\text{BT}}(r)| \leq 2\hat{\mathfrak{R}}_S(\ell \circ \mathcal{H}_{\text{pair}}) + 3M_B \sqrt{\frac{\log(2/\delta)}{n}} \tag{106}$$

In addition, we note that

$$g'_{x,y,y'}(a) = \omega_P - \sigma(a) \leq 1 \tag{107}$$

which implies that $g_{x,y,y'}$ is 1-Lipshitz. As a result, the Rademacher complexity of our loss must follows

$$\hat{\mathfrak{R}}_S(\ell \circ \mathcal{H}_{\text{pair}}) \leq \hat{\mathfrak{R}}_S(\mathcal{H}_{\text{pair}}) \cdot 1 \tag{108}$$

Substitute this back, we have

$$\sup_{r \in \mathcal{H}} \frac{1}{|Z|} |\mathcal{L}_{\text{BT}}(r) - \widehat{\mathcal{L}}_{\text{BT}}(r)| \leq 2\hat{\mathfrak{R}}_S(\mathcal{H}_{\text{pair}}) + 3M_B \sqrt{\frac{\log(2/\delta)}{n}} \tag{109}$$

and

$$\frac{1}{|Z|} \mathcal{L}_{\text{BT}}(\hat{r}) - \mathcal{L}_{\text{BT}}(r_{\text{opt}}) \leq 4\hat{\mathfrak{R}}_S(\mathcal{H}_{\text{pair}}) + 6M_B \sqrt{\frac{\log(2/\delta)}{n}} \tag{110}$$

**Step 5: Combine all the bounds together**
Here, we just combine everything together to get the final bound.

$$\mathbb{E}_{Q_{\text{pair}}}[(\Delta_{\hat{r}} - \Delta_{r^*})^2] \tag{111}$$

$$\leq 2\mathbb{E}_{Q_{\text{pair}}}[(\Delta_{\hat{r}} - \Delta_{r_{\text{opt}}})^2] + 2\mathbb{E}_{Q_{\text{pair}}}[(\Delta_{r_{\text{opt}}} - \Delta_{r^*})^2] \tag{112}$$

$$\leq \frac{4}{\lambda_{\text{conn}}} \mathbb{E}_{\tilde{P}}[(\Delta_{\hat{r}} - \Delta_{r_{\text{opt}}})^2] + 2\mathbb{E}_{Q_{\text{pair}}}[(\Delta_{r_{\text{opt}}} - \Delta_{r^*})^2] \tag{113}$$

$$\leq \frac{4}{\lambda_{\text{conn}}} (4((\frac{\epsilon_{\text{misspec}}}{c_B})^2 + \frac{1}{Zc_B}(\mathcal{L}_{\text{BT}}(\hat{r}) - \mathcal{L}_{\text{BT}}(r_{\text{opt}})))) + 2\mathbb{E}_{Q_{\text{pair}}}[(\Delta_{r_{\text{opt}}} - \Delta_{r^*})^2] \tag{114}$$

$$\leq \frac{4}{\lambda_{\text{conn}}} (4((\frac{\epsilon_{\text{misspec}}}{c_B})^2 + \frac{1}{c_B}(4\hat{\mathfrak{R}}_S(\mathcal{H}_{\text{pair}}) + 6M_B \sqrt{\frac{\log(2/\delta)}{n}}))) + 2\mathbb{E}_{Q_{\text{pair}}}[(\Delta_{r_{\text{opt}}} - \Delta_{r^*})^2] \tag{115}$$

$$= \frac{16}{\lambda_{\text{conn}}} ((\frac{\epsilon_{\text{misspec}}}{c_B})^2 + \frac{1}{c_B}(4\hat{\mathfrak{R}}_S(\mathcal{H}_{\text{pair}}) + 6M_B \sqrt{\frac{\log(2/\delta)}{n}})) + 2\mathbb{E}_{Q_{\text{pair}}}[(\Delta_{r_{\text{opt}}} - \Delta_{r^*})^2] \tag{116}$$

$$\lesssim \frac{1}{\lambda_{\text{conn}}} ((\frac{\epsilon_{\text{misspec}}}{c_B})^2 + \frac{1}{c_B}(\hat{\mathfrak{R}}_S(\mathcal{H}_{\text{pair}}) + M_B \sqrt{\frac{\log(2/\delta)}{n}})) + \mathbb{E}_{Q_{\text{pair}}}[(\Delta_{r_{\text{opt}}} - \Delta_{r^*})^2] \tag{117}$$

$$\square$$

**Corollary C.2** (Estimation error bound, realizable). *Let $r^*$ be the target score function, $\mathcal{H}$ be a hypothesis class such that $r^* \in \mathcal{H}$ and $\mathcal{H}$ is bounded by some constant $B$. Let $P$ be a triplet distribution that is BT-consistent with respect to $r^*$ and let $S$ be a set of $n$ samples drawn from $P$. Let $\hat{r}$ be the empirical risk minimizer of the BT learning objective on $S$ then there exists a constant $c_B, M_B > 0$ such that with probability at least $1 - \delta$ over $S$,*

$$\mathbb{E}_{Q_{pair}}[(\Delta_r - \Delta_{r^*})^2] \lesssim \frac{1}{\lambda_{conn}} (\frac{1}{c_B}(\hat{\mathfrak{R}}_S(\mathcal{H}_{pair}) + M_B \sqrt{\frac{\log(2/\delta)}{n}}))$$

when $\hat{\mathfrak{R}}_S$ is an empirical Rademacher complexity of the class $\mathcal{H}_{pair} := \{r(x,y) - r(x,y') : r \in \mathcal{H}\}$ and $\lambda_{conn}$ is the connectivity degree of $P$ with respect to $\mathcal{H}$ and $Q$.

*Proof.* In the realizable setting, we know that $r^* \in \mathcal{H}$ so we have $r_{opt} = r^*$. As a result $\epsilon^2_{misspec} = 0$ and $\mathbb{E}_{Q_{pair}}[(\Delta_{r_{opt}} - \Delta_r^*)^2] = 0$. Substituting these into the estimation error bound, we have the desired result. $\qquad\square$

**Proposition C.3** (Connectivity degree for tabular BT). *In classical setting of tabular BT, we only have one context $x$ and a finite number of responses $\mathcal{Y} = \{y_1, y_2, \ldots, y_m\}$ where each response has an associated score $\theta_i \in \mathbb{R}$ for $i = 1, 2, \ldots, m$. Our goal is to recover the $\theta$. Assuming that the evaluation distribution $Q$ is uniform over the response space and we restrict the hypothesis class to be $\mathcal{H} = \{\theta : \theta \in \mathbb{R}^m \sum_{i=1}^m \theta_i = 0\}$ such that the mean of the score is zero for identifiability then the connectivity degree is given by*

$$\lambda_{conn} = m\lambda_2(L) \tag{118}$$

*is the Fiedler value of the Laplacian matrix $L$ of the comparison graph. This graph is defined as a graph with $m$ nodes where node $i$ represents the response $y_i$ and the edge weight between node $i$ and node $j$ is the probability that $y_i$ and $y_j$ are compared.*

*Proof.* From the definition of the connectivity degree,

$$\lambda_{conn} = \inf_{f,g \in \mathcal{H}} \frac{\mathbb{E}_{\widetilde{P}}[(\Delta f - \Delta g)^2]}{\widetilde{\mathrm{Var}}_Q[(f - g)]} \tag{119}$$

Our first observation is that $\mathcal{H} - \mathcal{H} = \{\theta - \theta' : \theta, \theta' \in \mathcal{H}\} = \mathcal{H}$. This holds since our $\theta$ is any real value vector with the mean of zero, so the difference between any two score functions would still belong to the same class. As a result, we can write the connectivity as

$$\lambda_{conn} = \inf_{\theta \in \mathbb{R}^m, \theta^\top \mathbf{1} = 0} \frac{\mathbb{E}_{\widetilde{P}}[\Delta\theta^2]}{\widetilde{\mathrm{Var}}_Q[\theta]} \tag{120}$$

Now, let's unpack each term at a time. For the numerator, we have

$$\mathbb{E}_{\widetilde{P}}[\Delta\theta^2] = \mathbb{E}_{y_i,y_j \sim \widetilde{P}}[(\theta_i - \theta_j)^2] = \theta^\top L\theta \tag{121}$$

when $L$ is the Laplacian matrix of the comparison graph where the edge weight between node $i$ and node $j$ is given by $\widetilde{P}_{ij}$ the probability for which $y_i$ and $y_j$ are compared.

For the second term, we have

$$\widetilde{\mathrm{Var}}_Q[\theta] = \mathbb{E}_{x \sim Q_x} \mathrm{Var}_{y \sim Q_y(y|x)}[\theta] \tag{122}$$

Since, there is only one $x$, we are left with just the variance of the score function.

$$\widetilde{\mathrm{Var}}_Q[\theta] = \mathrm{Var}_{y \sim Q_y}[\theta] = \mathbb{E}_{Q_y}[\theta^2] - (\mathbb{E}_{Q_y}[\theta])^2 \tag{123}$$

From our assumption, $Q$ is a uniform distribution over the response space, and $\theta$ has mean of zero, we can conclude that $\mathbb{E}_{Q_y}[\theta] = \frac{1}{m}\sum_{i=1}^m \theta_i = 0$ and

$$\widetilde{\mathrm{Var}}_Q[\theta] = \mathbb{E}_{Q_y}[\theta^2] = \frac{1}{m}\theta^\top L\theta \tag{124}$$

Therefore,

$$\lambda_{conn} = \inf_{\theta \in \mathbb{R}^m, \theta^\top \mathbf{1} = 0} \frac{\theta^\top L\theta}{\frac{1}{m}\theta^\top \theta} = m\lambda_2(L) \tag{125}$$

where $\lambda_2(L)$ is the second smallest eigenvalue of the Laplacian matrix $L$. This holds from the Rayleigh quotient. $\qquad\square$

Our proposed connectivity degree recovers the Fiedler value which appears in the prior work that provide an estimation bound for this classical tabular BT setting (Shah et al., 2016).

**Proposition C.4** (Connectivity degree for linear BT). *In the setting of linear BT, we have a hypothesis class of $\mathcal{H} = \{r : r(x,y) = w^\top \phi(x,y), w \in \mathbb{R}^d\}$ where $\phi(x,y)$ is a feature map. Let Q be the evaluation distribution and assume that the feature map is centered around Q, that is for any context x, we have $\mathbb{E}_{y \sim Q_y(y|x)}[\phi(x,y)] = 0$, then the connectivity degree is given by*

$$\lambda_{conn} = \lambda_{\min}(\Sigma_Q^{-1/2}\Sigma_P\Sigma_Q^{-1/2}) \tag{126}$$

*is the smallest eigenvalue of the matrix $\Sigma_Q^{-1/2}\Sigma_P\Sigma_Q^{-1/2}$ where $\Sigma_P \in \mathbb{R}^{d \times d}$ is defined as*

$$\Sigma_P = \mathbb{E}_{(x,y,y') \sim \widetilde{P}}[(\phi(x,y) - \phi(x,y'))(\phi(x,y) - \phi(x,y'))^\top] \tag{127}$$

*and the matrix $\Sigma_Q \in \mathbb{R}^{d \times d}$ is defined as*

$$\Sigma_Q = \mathbb{E}_{(x,y) \sim Q}[\phi(x,y)\phi(x,y)^\top] \tag{128}$$

*Proof.* Similar to the proof of the connectivity degree for tabular BT, we first note that $\mathcal{H} - \mathcal{H} = \{r - r' : r, r' \in \mathcal{H}\} = \mathcal{H}$. This holds since our $r$ is any linear function of the feature map, so the difference between any two score functions would still be a linear function of the feature map. As a result, we can write the connectivity as

$$\lambda_{\mathrm{conn}} = \inf_{w \in \mathbb{R}^d} \frac{\mathbb{E}_{\widetilde{P}}[(\Delta w^\top \phi)^2]}{\widetilde{\mathrm{Var}}_Q[w^\top \phi]} \tag{129}$$

For the numerator, we have

$$\mathbb{E}_{\widetilde{P}}[(\Delta w^\top \phi)^2] = \mathbb{E}_{(x,y,y') \sim \widetilde{P}}[(w^\top \phi(x,y) - w^\top \phi(x,y'))^2] \tag{130}$$

$$= \mathbb{E}_{(x,y,y') \sim \widetilde{P}}[(w^\top(\phi(x,y) - \phi(x,y')))^2] \tag{131}$$

$$= \mathbb{E}_{(x,y,y') \sim \widetilde{P}}[(w^\top(\phi(x,y) - \phi(x,y')))(\phi(x,y) - \phi(x,y'))^\top w] \tag{132}$$

$$= w^\top \mathbb{E}_{(x,y,y') \sim \widetilde{P}}[(\phi(x,y) - \phi(x,y'))(\phi(x,y) - \phi(x,y'))^\top] w \tag{133}$$

$$= w^\top \Sigma_P w \tag{134}$$

For the denominator, we have

$$\widetilde{\mathrm{Var}}_Q[w^\top \phi] = \mathbb{E}_{x \sim Q_x} \mathrm{Var}_{y \sim Q_y(y|x)}[w^\top \phi(x,y)] \tag{135}$$

$$\mathrm{Var}_{y \sim Q_y(y|x)}[w^\top \phi(x,y)] = \mathbb{E}_{y \sim Q_y(y|x)}[(w^\top \phi(x,y))^2] - (\mathbb{E}_{y \sim Q_y(y|x)}[w^\top \phi(x,y)])^2 \tag{136}$$

$$= \mathbb{E}_{y \sim Q_y(y|x)}[w^\top \phi(x,y)\phi(x,y)^\top w] - (w^\top \mathbb{E}_{y \sim Q_y(y|x)}[\phi(x,y)])^2 \tag{137}$$

$$= w^\top \mathbb{E}_{y \sim Q_y(y|x)}[\phi(x,y)\phi(x,y)^\top] w - (w^\top \mathbb{E}_{y \sim Q_y(y|x)}[\phi(x,y)])^2 \tag{138}$$

$$= w^\top \mathbb{E}_{y \sim Q_y(y|x)}[\phi(x,y)\phi(x,y)^\top] w \tag{139}$$

The last term is zero since $\mathbb{E}_{y \sim Q_y(y|x)}[\phi(x,y)] = 0$ by our assumption. Therefore, we have

$$\widetilde{\mathrm{Var}}_Q[w^\top \phi] = \mathbb{E}_{x \sim Q_x} w^\top \mathbb{E}_{y \sim Q_y(y|x)}[\phi(x,y)\phi(x,y)^\top] w \tag{140}$$

$$= w^\top \mathbb{E}_{x \sim Q_x} \mathbb{E}_{y \sim Q_y(y|x)}[\phi(x,y)\phi(x,y)^\top] w \tag{141}$$

$$= w^\top \Sigma_Q w \tag{142}$$

Putting everything together, we have

$$\lambda_{\mathrm{conn}} = \inf_{w \in \mathbb{R}^d} \frac{w^\top \Sigma_P w}{w^\top \Sigma_Q w} = \lambda_{\min}(\Sigma_Q^{-1/2}\Sigma_P\Sigma_Q^{-1/2}) \tag{143}$$

where $\lambda_{\min}(\cdot)$ is the smallest eigenvalue of the matrix. This holds from the Rayleigh quotient. $\qquad \square$

Our result compliments the result in (Zhu et al., 2023), which provide a bound for recovering the parameter $w$ under $\|\cdot\|_{\Sigma_P}$. Our result shows how to transfer to the evaluation under the test distribution $Q$. This $\Sigma_P$ is also related to the Fisher information matrix that is used on active reward learning in (Shen et al., 2025). The key difference is that our $\Sigma_P$ is independent of the current hypothesis $w$ and is only dependent on the comparison distribution $\widetilde{P}$ and the hypothesis class $\mathcal{H}$ while Fisher information in (Shen et al., 2025) is dependent on the current hypothesis $w$.

**Theorem 6.8** (Accuracy bound, realizable case). *Let $\hat{r}$ be the empirical risk minizer of the BT learning objective, then there exists a constant $D > 0$ such that with probability at least $1 - \delta$,*

$$
\mathrm{Acc}_Q(\hat{r}) \geq \sup_{k>0} \Big[ \underbrace{\Pr_Q(|\Delta_{r^*}| \geq k)}_{margin} - \underbrace{D\frac{\mathrm{Comp}(\mathcal{H}, \delta)}{k^2 \lambda_{conn}}}_{connectivity} \Big] \tag{25}
$$

*where the complexity term is defined as in right hand side of the Theorem 6.7.*

*Proof.* We will prove this using Markov's inequality and some algebra. Recall from Theorem 6.4 that

$$
\mathrm{Acc}_Q(\hat{r}) \geq \Pr_{Q_{\mathrm{pair}}} \left( |\Delta_{\hat{r}} - \Delta_{r^*}| \leq |\Delta_{r*}| \right) \tag{144}
$$

For any $k > 0$, the following hold

$$
\Pr_{Q_{\mathrm{pair}}} \left( |\Delta_{\hat{r}} - \Delta_{r*}| \leq |\Delta_{r*}| \right) \geq \Pr_{Q_{\mathrm{pair}}} \left( |\Delta_{\hat{r}} - \Delta_{r*}| < k, |\Delta_{r*}| \geq k \right) \tag{145}
$$

$$
\geq \Pr_{Q_{\mathrm{pair}}} \left( |\Delta_{\hat{r}} - \Delta_{r*}| < k \right) + \Pr_{Q_{\mathrm{pair}}} \left( |\Delta_{r*}| \geq k \right) - 1 \tag{146}
$$

$$
= \Pr_{Q_{\mathrm{pair}}} \left( |\Delta_{r*}| \geq k \right) - \Pr_{Q_{\mathrm{pair}}} \left( |\Delta_{\hat{r}} - \Delta_{r*}| \geq k \right) \tag{147}
$$

This is true from the identity that $\Pr(A \cap B) \geq \Pr(A) + \Pr(B) - 1$. Now, from Markov's inequality, we know that

$$
\Pr_{Q_{\mathrm{pair}}} \left( |\Delta_{\hat{r}} - \Delta_{r*}| \geq k \right) = \Pr_{Q_{\mathrm{pair}}} \left( |\Delta_{\hat{r}} - \Delta_{r*}|^2 \geq k^2 \right) \tag{148}
$$

$$
\leq \frac{\mathbb{E}_{Q_{\mathrm{pair}}}[(\Delta_{\hat{r}} - \Delta_{r*})^2]}{k^2} \tag{149}
$$

$$
\leq \frac{c(\frac{1}{c_B}(\hat{\mathfrak{R}}_S(\mathcal{H}_{\mathrm{pair}}) + M_B \sqrt{\frac{\log(2/\delta)}{n}}))}{\lambda_{\mathrm{conn}} k^2} \tag{150}
$$

$$
=: \frac{c\,\mathrm{Complexity}(\mathcal{H}, \delta)}{k^2 \lambda_{\mathrm{conn}}} \tag{151}
$$

when $c > 0$ is a constant. The final line follows from Theorem 6.7 which holds with probability at least $1 - \delta$. Substituting this back, we have

$$
\Pr_{Q_{\mathrm{pair}}} \left( |\Delta_r - \Delta_{r*}| \leq |\Delta_{r*}| \right) \geq \Pr_{Q_{\mathrm{pair}}} \left( |\Delta_{r*}| \geq k \right) - \frac{c\,\mathrm{Complexity}(\mathcal{H}, \delta)}{k^2 \lambda_{\mathrm{conn}}} \tag{152}
$$

Taking the supremum over all $k > 0$ for the right hand side and we have the desired result. $\square$

# D. Additional Experiments

## D.1. Accuracy on pairs with small margin

From Figure 4a and 4b, we can see that the accuracy gap between $r^*_{\text{rank}}$ and $r^*$ is larger than the overall accuracy. It takes much less data to achieve the same accuracy for $r^*_{\text{rank}}$ than $r^*$.

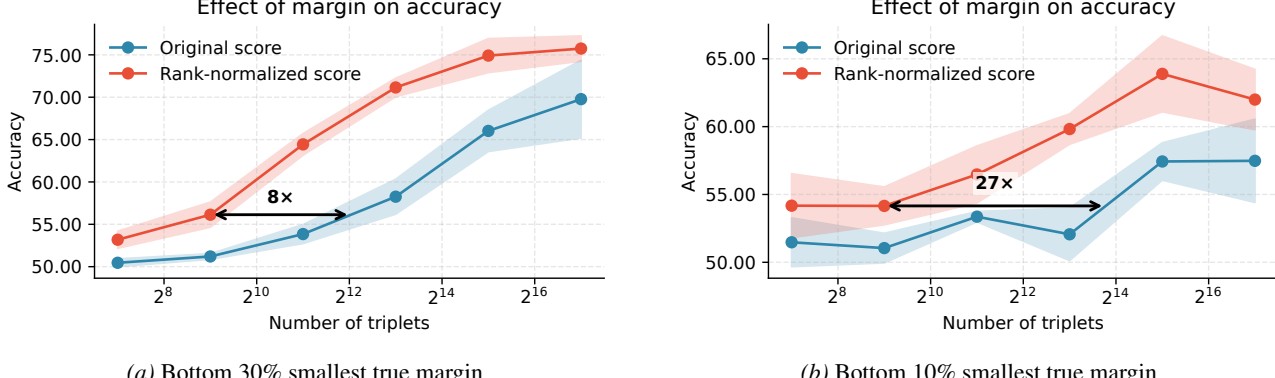

*(a)* Bottom 30% smallest true margin

*(b)* Bottom 10% smallest true margin

*Figure 4.* Effect of score margin on learning efficiency on pairs with the smallest true margin.

## D.2. Connectivity Degree

We provide more details on how we calculate the connectivity degree. The connectivity degree is defined as

$$\lambda_{\text{conn}}(P, Q; \mathcal{H}) = \inf_{f,g \in \mathcal{H}} \frac{\mathbb{E}_{\widetilde{P}}[(\Delta f - \Delta g)^2]}{\widetilde{\text{Var}}_Q[(f - g)]} \tag{153}$$

Since we are in the finite setting with $\mathcal{X} = \{x_1, \ldots, x_m\}$, let $\tilde{P}_{ijk} = \widetilde{P}(x_i, \{x_j, x_k\})$ for convenience, then the numerator is given by

$$\mathbb{E}_{\widetilde{P}}[(\Delta f - \Delta g)^2] = \sum_{i,j,k} \tilde{P}_{ijk}(\Delta f(x_i, x_j, x_k) - \Delta g(x_i, x_j, x_k))^2. \tag{154}$$

Our test distribution $Q$ is a uniform distribution over the response space, so the variance is given by

$$\widetilde{\text{Var}}_Q[(f - g)] = \frac{1}{m} \sum_{i=1}^{m} \left[ \frac{1}{m} \sum_{j=1}^{m} (f(x_i, x_j) - g(x_i, x_j))^2 - \left( \frac{1}{m} \sum_{j=1}^{m} (f(x_i, x_j) - g(x_i, x_j)) \right)^2 \right]. \tag{155}$$

Since we have access to $\widetilde{P}$ and all possible value of $x_i$, we can compute each term in the numerator and denominator for any function $f$ and $g$. To estimate the connectivity degree, we initialize the score function $f$ and $g$ in the form of cosine similarity between two hidden representations $h(x)$ and $h(y)$ parameterized by two-layer neural networks.

$$f(x, y) = \text{cosine-similarity}(h_f(x), h_f(y)), \tag{156}$$

$$g(x, y) = \text{cosine-similarity}(h_g(x), h_g(y)), \tag{157}$$

We then jointly train $f$ and $g$ to minimize the term in the connectivity degree definition, using gradient descent.

To optimize $p^-$ that maximizes the connectivity degree, we can turn this into a minimax problem. For any $p^-$, we can compute the corresponding $p^+$ that is BT-consistent and as a result can compute the corresponding comparison distribution $\widetilde{P}$. Our optimization objective becomes

$$\max_{p^-} \lambda_{\text{conn}}(P(p^-), Q; \mathcal{H}) \tag{158}$$

*Table 3.* Stability of the estimated connectivity degree from the minimax optimization. For each value of $\alpha$, the estimation is repeated 10 times and aggregated across 5 random seeds.

| $\alpha$ | Mean | Std. | Relative std. |
|---|---|---|---|
| $-16.0$ | 0.0333 | 0.0260 | 95.9% |
| $-8.0$ | 0.2159 | 0.0461 | 33.2% |
| $-4.0$ | 0.9569 | 0.0522 | 7.3% |
| $-2.0$ | 2.2040 | 0.0292 | 1.5% |
| $-1.0$ | 3.3330 | 0.0130 | 0.4% |
| $-0.5$ | 3.9672 | 0.0003 | 0.01% |
| 0.0 | 3.2801 | 0.0075 | 0.2% |
| 0.5 | 2.6066 | 0.0191 | 0.8% |
| 1.0 | 2.0599 | 0.0296 | 1.6% |
| 2.0 | 1.2605 | 0.0267 | 2.5% |
| 4.0 | 0.4888 | 0.0386 | 9.8% |
| 8.0 | 0.0850 | 0.0086 | 17.8% |
| 16.0 | 0.0060 | 0.0020 | 44.1% |

which leads to the following optimization problem:

$$\max_{p^-} \inf_{f,g \in \mathcal{H}} \frac{\mathbb{E}_{\widetilde{P}(p^-)}[(\Delta f - \Delta g)^2]}{\widetilde{\mathrm{Var}}_Q[(f-g)]} \tag{159}$$

What we can do is to initialize $p^-$ as a uniform distribution and $f, g$ in the similar form as before, then we can apply alternating gradient descent ascent to update $p^-$ and $f, g$.

**Stability of the estimated connectivity degree.** We provide a stability analysis for the connectivity degree estimated by the minimax optimization in Section 7.3. For each value of $\alpha$, we repeated the estimation 10 times and aggregated the results across 5 random seeds. Table 3 reports the average mean, average standard deviation, and average relative standard deviation.

Overall, the estimate is fairly stable in the main regime of interest. For $\alpha \in [-4, 4]$, the relative standard deviation is at most about 10%, and is below 3% for most values in this range. For more extreme values of $\alpha$, the relative variation becomes larger, but this occurs precisely when the connectivity degree itself is already very small. In these cases, the absolute standard deviation remains small; for example, it is 0.0260 at $\alpha = -16$ and 0.0020 at $\alpha = 16$. These results suggest that, although the minimax estimate is stochastic, it is sufficiently stable to support the trends reported in Section 7.3.

### D.3. Connectivity Degree General Setting

When we have general test distribution $Q'$ over $\mathcal{X} \times \mathcal{Y} \times \mathcal{Y}$. The connectivity degree is generalized to

$$\lambda_{\mathrm{conn}}(P, Q'; \mathcal{H}) = \inf_{f,g \in H} \frac{\mathbb{E}_{\widetilde{P}}[(\Delta f - \Delta g)^2]}{\mathbb{E}_{Q'}[(\Delta f - \Delta g)^2]} \tag{160}$$

This general quantity is a worst-case ratio between disagreement on the training and test distribution. When $Q' = P$, this quantity is 1. Previously, we focus on the test distribution with the product form $Q'(x, y, y') = Q_x(x)Q(y \mid x)Q(y' \mid x)$. Under this choice, we evaluate over all independent pairs of responses drawn from $Q(\cdot \mid x)$ rather than only over some selective pairs. This gives the quantity a connectivity interpretation: the ratio asks whether the training distribution captures enough pairwise information to control disagreement over the full response space over independent pairs.

Across datasets, connectivity is larger when the training comparisons better cover the pairwise distinctions that matter under the target evaluation distribution. Across models, expanding the hypothesis class can decrease connectivity because the infimum is taken over a richer set of functions. This is precisely why connectivity is informative: it captures the interaction between the data distribution and the model class. In practice, this motivates estimating connectivity using a tractable hypothesis class, such as linear models, as a proxy for more complex classes: if connectivity is poor even for the linear class, it will be no larger for richer classes.

In the experiment in Section 7.6, we compute the connectivity degree for a class of linear reward heads over the embeddings

from `Llama-3.1-8B-Instruct`. For this linear setting, the connectivity degree is simply the smallest eigenvalue of the covariance matrix.

**Proposition D.1** (Connectivity degree for linear BT with general test distribution). *In the setting of linear BT, let*

$$\mathcal{H} = \{r : r(x,y) = w^\top \phi(x,y), \ w \in \mathbb{R}^d\}, \tag{161}$$

*where $\phi(x,y)$ is a feature map. Let $P$ be the training comparison distribution and let $Q'$ be a general test distribution over $\mathcal{X} \times \mathcal{Y} \times \mathcal{Y}$. Define*

$$\Sigma_P = \mathbb{E}_{(x,y,y')\sim P}\left[(\phi(x,y) - \phi(x,y'))(\phi(x,y) - \phi(x,y'))^\top\right], \tag{162}$$

*and*

$$\Sigma_{Q'} = \mathbb{E}_{(x,y,y')\sim Q'}\left[(\phi(x,y) - \phi(x,y'))(\phi(x,y) - \phi(x,y'))^\top\right]. \tag{163}$$

*Then the connectivity degree is given by*

$$\lambda_{conn}(P,Q';\mathcal{H}) = \lambda_{\min}\left(\Sigma_{Q'}^{-1/2}\Sigma_P\Sigma_{Q'}^{-1/2}\right), \tag{164}$$

*where $\lambda_{\min}(\cdot)$ denotes the smallest eigenvalue.*

*Proof.* First note that

$$\mathcal{H} - \mathcal{H} = \{r - r' : r, r' \in \mathcal{H}\} = \mathcal{H}, \tag{165}$$

since the difference between two linear reward functions is still linear in $\phi$. Therefore, it suffices to consider functions of the form $r_w(x,y) = w^\top \phi(x,y)$.

By definition of connectivity under a general test comparison distribution,

$$\lambda_{\text{conn}}(P,Q';\mathcal{H}) = \inf_{w\in\mathbb{R}^d} \frac{\mathbb{E}_{(x,y,y')\sim P}\left[\left(w^\top\phi(x,y) - w^\top\phi(x,y')\right)^2\right]}{\mathbb{E}_{(x,y,y')\sim Q'}\left[\left(w^\top\phi(x,y) - w^\top\phi(x,y')\right)^2\right]}. \tag{166}$$

For the numerator, we have

$$\begin{aligned}
\mathbb{E}_{(x,y,y')\sim P}&\left[\left(w^\top\phi(x,y) - w^\top\phi(x,y')\right)^2\right] \\
&= \mathbb{E}_{(x,y,y')\sim P}\left[\left(w^\top(\phi(x,y) - \phi(x,y'))\right)^2\right] \\
&= w^\top\Sigma_P w.
\end{aligned} \tag{167}$$

Similarly, for the denominator,

$$\begin{aligned}
\mathbb{E}_{(x,y,y')\sim Q'}&\left[\left(w^\top\phi(x,y) - w^\top\phi(x,y')\right)^2\right] \\
&= \mathbb{E}_{(x,y,y')\sim Q'}\left[\left(w^\top(\phi(x,y) - \phi(x,y'))\right)^2\right] \\
&= w^\top\Sigma_{Q'} w.
\end{aligned} \tag{168}$$

Thus,

$$\lambda_{\text{conn}}(P,Q';\mathcal{H}) = \inf_{w\neq 0} \frac{w^\top\Sigma_P w}{w^\top\Sigma_{Q'} w}. \tag{169}$$

By the Rayleigh quotient for generalized eigenvalues,

$$\inf_{w\neq 0} \frac{w^\top\Sigma_P w}{w^\top\Sigma_{Q'} w} = \lambda_{\min}\left(\Sigma_{Q'}^{-1/2}\Sigma_P\Sigma_{Q'}^{-1/2}\right). \tag{170}$$

This proves the claim. □

## D.4. Synthetic non-BT CPRD experiment

We construct a synthetic non-BT CPRD whose BT projection has the same target score $r^*$ as the original BT-consistent distribution. Let $P_{\mathrm{BT}}$ denote the original BT-consistent triplet distribution. For each unordered pair $\{y_i, y_j\}$, define its unordered comparison mass

$$M_{\mathrm{BT}}(x, i, j) := P_{\mathrm{BT}}(x, y_i, y_j) + P_{\mathrm{BT}}(x, y_j, y_i). \tag{171}$$

The induced comparison distribution $\widetilde{P}$ is proportional to this mass:

$$\widetilde{P}(x, \{y_i, y_j\}) = \frac{M_{\mathrm{BT}}(x, i, j)}{Z}, \tag{172}$$

where $Z$ is the normalizing constant from Theorem 5.2. The CPRD of $P_{\mathrm{BT}}$ is

$$\omega_{P_{\mathrm{BT}}}(y_i \succ y_j \mid x) = \sigma(r^*(x, y_i) - r^*(x, y_j)). \tag{173}$$

**Perturbing the oriented triplet mass.** We construct $P_{\mathrm{nonBT}}$ by perturbing the two orientations of each unordered pair in opposite directions. Let $g_x(i, j)$ be an antisymmetric perturbation,

$$g_x(i, j) = -g_x(j, i), \tag{174}$$

supported only on pairs with $M_{\mathrm{BT}}(x, i, j) > 0$. We define

$$P_{\mathrm{nonBT}}(x, y_i, y_j) = P_{\mathrm{BT}}(x, y_i, y_j) + \varepsilon g_x(i, j), \tag{175}$$

and equivalently

$$P_{\mathrm{nonBT}}(x, y_j, y_i) = P_{\mathrm{BT}}(x, y_j, y_i) - \varepsilon g_x(i, j). \tag{176}$$

Thus the unordered comparison mass is unchanged:

$$P_{\mathrm{nonBT}}(x, y_i, y_j) + P_{\mathrm{nonBT}}(x, y_j, y_i) = P_{\mathrm{BT}}(x, y_i, y_j) + P_{\mathrm{BT}}(x, y_j, y_i) = M_{\mathrm{BT}}(x, i, j). \tag{177}$$

Therefore $P_{\mathrm{BT}}$ and $P_{\mathrm{nonBT}}$ induce the same comparison distribution $\widetilde{P}$. Only the directional split within each unordered pair changes. By the definition of the CPRD,

$$\omega_P(y_i \succ y_j \mid x) = \frac{P(x, y_i, y_j)}{P(x, y_i, y_j) + P(x, y_j, y_i)}. \tag{178}$$

Therefore,

$$\omega_{P_{\mathrm{nonBT}}}(y_i \succ y_j \mid x) = \frac{P_{\mathrm{BT}}(x, y_i, y_j) + \varepsilon g_x(i, j)}{M_{\mathrm{BT}}(x, i, j)} \tag{179}$$

$$= \omega_{P_{\mathrm{BT}}}(y_i \succ y_j \mid x) + \varepsilon h_x(i, j), \tag{180}$$

where

$$h_x(i, j) := \frac{g_x(i, j)}{M_{\mathrm{BT}}(x, i, j)}. \tag{181}$$

**Condition for preserving the BT projection.** By Theorem 5.2, the population BT objective can be written, up to constants independent of $r$, as a cross-entropy projection of the CPRD onto the BT family under the comparison distribution $\widetilde{P}$. Thus, for $P_{\mathrm{nonBT}}$, the relevant objective is

$$\mathcal{L}_{\mathrm{BT}}^{\mathrm{nonBT}}(r) = \sum_{x, i < j} \widetilde{P}(x, \{y_i, y_j\}) \, \mathrm{CE}(\mathrm{Bern}(\omega_{P_{\mathrm{nonBT}}}(y_i \succ y_j \mid x)), \mathrm{Bern}(\sigma(r(x, y_i) - r(x, y_j)))). \tag{182}$$

We now derive the condition under which $r^*$ remains the BT projection target. Fix $x$ and write $r_i = r(x, y_i)$, $\omega_{ij} = \omega_{P_{\mathrm{nonBT}}}(y_i \succ y_j \mid x)$, and $\Delta_{ij} = r_i - r_j$. For one unordered pair $\{y_i, y_j\}$, the Bernoulli cross-entropy term is

$$\ell_{ij}(r) = -\omega_{ij} \log \sigma(\Delta_{ij}) - (1 - \omega_{ij}) \log \sigma(-\Delta_{ij}). \tag{183}$$

From equation (81), we have

$$\frac{\partial \ell_{ij}(r)}{\partial r_i} = -\omega_{ij}(1 - \sigma(\Delta_{ij})) + (1 - \omega_{ij})\sigma(\Delta_{ij}) \tag{184}$$

$$= \sigma(\Delta_{ij}) - \omega_{ij}. \tag{185}$$

Applying this to all pairs involving $y_i$ gives

$$\frac{\partial \mathcal{L}_{\mathrm{BT}}^{\mathrm{nonBT}}(r)}{\partial r(x, y_i)} = \sum_{j \neq i} \widetilde{P}(x, \{y_i, y_j\}) \left[ \sigma(r_i - r_j) - \omega_{P_{\mathrm{nonBT}}}(y_i \succ y_j \mid x) \right]. \tag{186}$$

Now evaluate the gradient at $r = r^*$. Since $P_{\mathrm{BT}}$ is BT-consistent,

$$\omega_{P_{\mathrm{BT}}}(y_i \succ y_j \mid x) = \sigma(r_i^* - r_j^*). \tag{187}$$

Using the induced CPRD perturbation,

$$\omega_{P_{\mathrm{nonBT}}}(y_i \succ y_j \mid x) = \omega_{P_{\mathrm{BT}}}(y_i \succ y_j \mid x) + \varepsilon h_x(i, j), \tag{188}$$

we obtain

$$\frac{\partial \mathcal{L}_{\mathrm{BT}}^{\mathrm{nonBT}}(r^*)}{\partial r(x, y_i)} = \sum_{j \neq i} \widetilde{P}(x, \{y_i, y_j\}) \left[ \sigma(r_i^* - r_j^*) - \omega_{P_{\mathrm{nonBT}}}(y_i \succ y_j \mid x) \right] \tag{189}$$

$$= \sum_{j \neq i} \widetilde{P}(x, \{y_i, y_j\}) \left[ \omega_{P_{\mathrm{BT}}}(y_i \succ y_j \mid x) - \omega_{P_{\mathrm{BT}}}(y_i \succ y_j \mid x) - \varepsilon h_x(i, j) \right] \tag{190}$$

$$= -\varepsilon \sum_{j \neq i} \widetilde{P}(x, \{y_i, y_j\}) h_x(i, j). \tag{191}$$

Since

$$\widetilde{P}(x, \{y_i, y_j\}) h_x(i, j) = \frac{M_{\mathrm{BT}}(x, i, j)}{Z} \cdot \frac{g_x(i, j)}{M_{\mathrm{BT}}(x, i, j)} = \frac{g_x(i, j)}{Z}, \tag{192}$$

the gradient shift is

$$\frac{\partial \mathcal{L}_{\mathrm{BT}}^{\mathrm{nonBT}}(r^*)}{\partial r(x, y_i)} = -\frac{\varepsilon}{Z} \sum_{j \neq i} g_x(i, j). \tag{193}$$

Therefore, a sufficient condition for preserving the BT projection target is the zero-flow condition

$$\sum_{j \neq i} g_x(i, j) = 0 \qquad \text{for all } x, i. \tag{194}$$

Equivalently,

$$\sum_{j \neq i} \widetilde{P}(x, \{y_i, y_j\}) h_x(i, j) = 0 \qquad \text{for all } x, i. \tag{195}$$

Under this condition, $r^*$ remains a stationary point of the BT projection objective. Since the BT cross-entropy objective is convex in the tabular scores, up to the usual additive shift fixed by the identifiability convention, $r^*$ remains the BT projection target for $P_{\mathrm{nonBT}}$.

**Choice of the perturbation $g_x$.** We choose $g_x$ to be a sum of signed circulations on directed 3-cycles. For a cycle $(a, b, c)$, define

$$g_x^{(a,b,c)}(i, j) = \mathbf{1}\{(i, j) \in \{(a, b), (b, c), (c, a)\}\} - \mathbf{1}\{(i, j) \in \{(b, a), (c, b), (a, c)\}\}. \tag{196}$$

This perturbation is antisymmetric. It also has zero row sum:

$$\sum_{j \neq i} g_x^{(a,b,c)}(i, j) = 0 \qquad \text{for all } i. \tag{197}$$

Because each cycle has zero row sum, their sum also satisfies

$$\sum_{j \neq i} g_x(i, j) = 0 \qquad \text{for all } x, i. \tag{198}$$

Thus the perturbation preserves the BT projection target by (194). In our implementation, we sample valid 3-cycles from the support of the comparison graph and randomly flip their orientations. We use at most 20 cycles per context. Validation and test sizes are both 2048, and final results are averaged over five random seeds.

