# OpenReview forum: "What Does Preference Learning Recover from Pairwise Comparison Data?"
_ICML.cc/2026/Conference — ICML 2026 regular_

### Official Review · Reviewer_Fxxf · 2026-03-07

**Soundness:** 3
**Presentation:** 3
**Significance:** 3
**Originality:** 2
**Overall Recommendation:** 4
**Confidence:** 4

**Summary:**

The paper presents a formalization of information in preference data, characterizes when BT modeling is appropriate, demonstrates what BT learning recovers from the data, and how the margin and connectivity affect learning. They also provide experimental results on 2-layer neural networks and synthetic data.

**Compliance With Llm Reviewing Policy:**

Affirmed.

**Final Justification:**

Thank you for the response. The additional experimental results were helpful in showing that, despite the bounds not being tight, connectivity does have a meaningful influence on model performance. This strengthens the sample complexity analysis. The additional clarifications on the practical impact, KL perspective, and connectivity are helpful, and have addressed my concerns. Including these in the revision would strengthen the paper. However, as noted by the authors, the paper is a first step, and I also think that while the results are clear and novel, the impact remains limited. I will raise my score to a 4.

**Key Questions For Authors:**

- Would it be possible to provide experimental results on language models or discuss how these results would scale to larger models?
- Could you provide some intuition on connectivity and how it might vary across datasets or models? I'm uncertain about how much we learn from connectivity, especially as it also seems very difficult to estimate as datasets and models grow.
- Could you share your thoughts on what the KL projection perspective implies for preference learning? How might it affect common beliefs about preference learning?

**Limitations:**

Yes

**Strengths And Weaknesses:**

Strengths:

- The analysis is rigorous and provides a formalization of the relationship between BT modeling and preference learning. The interpretation of the BT objective as performing a KL projection provides a novel and interesting perspective.
- The key results can be verified through experiments.

Weaknesses:

- The theorems correspond to an idealization of preference learning in practice. The results utilize abstractions or idealizations of model architecture, the structure of language data, and the preference distribution, making it difficult to apply the results to a given practical setting.
- While the KL projection perspective is interesting, it is unclear what conclusions to draw from this idea. Further discussion on what this implies for preference learning and the role of the data distribution would be helpful.
- The bounds on error and accuracy are likely vacuous, especially for large language models where preference learning is often applied, and as a result, it seems unclear to what extent the trends between the bound and margin or connectivity hold in practice. Experiments beyond two-layer neural networks would be particularly helpful in verifying the utility of the results.
- The intuition that a larger reward margin leads to easier learning is one that has often come up in analyses of preference learning and more generally in binary classification. Furthermore, connectivity seems to primarily be dependent on the hypothesis class and seems difficult to compute for larger models. The paper would be strengthened by providing a discussion on how these results might generalize as the model scale grows. It would also be strengthened by a more in-depth analysis of the error bound that goes beyond these two factors or provides more direct implications.

---

> ### Author Rebuttal · Authors · 2026-03-29
>
> Thank you for pressing on the practical implications.
>
> **1.Practical impact.**
>
> We view the paper as providing a principled understanding of preference learning from triplet data: what object the data encodes (the CPRD), when BT is a suitable model for that object, what BT learning recovers under misspecification, and how sample complexity depends on the data distribution through margin and connectivity. Although the paper does not yet provide a full procedure for improving preference data generation, it identifies the key quantities to optimize, namely margin and connectivity, which govern statistical efficiency. The paper also provides a useful heuristic for when BT is a reasonable modeling assumption through our characterization of positive–negative conditional independence. If responses are highly dependent, BT may not be the right model. We view this as a first step toward more actionable methods for designing preference data, much as formal analyses in other areas of machine learning have helped guide method development. We agree that the paper should make this role clearer.
>
> **2. Would it be possible to provide experimental results on language models.**
>
> We train a linear reward head on frozen embeddings using 10K preference pairs from HH-RLHF, PKU-SafeRLHF, SHP, and UltraFeedback, and evaluate on an UltraFeedback test set relabeled using Skywork-Reward-V2-Llama-3.1-8B. This lets us compute both the margin and a test-distribution variant of the connectivity degree in a controlled setting. The table below  provides results averaging over 5 random seeds with standard errors.
>
>
> | Dataset | Margin | Connectivity degree $(×10^{-4})$ | Accuracy |
> |---|---:|---:|---:|
> | HH-RLHF | 10.766 ± 0.036 | 3.30 ± 0.05 | 74.020 ± 0.339 |
> | PKU-SafeRLHF | 10.527 ± 0.017 | 0.85 ± 0.05 | 74.790 ± 0.545 |
> | SHP | 8.833 ± 0.026 | 15.14 ± 0.29 | 78.380 ± 0.244 |
> | Ultrafeedback | 11.255 ± 0.029 | 8.96 ± 0.16 | 82.170 ± 0.129 |
>
> **Takeaways.**
> - Connectivity matters: HH-RLHF and PKU-SafeRLHF have reasonable margins but very low connectivity, and both underperform SHP. The particularly low connectivity of HH-RLHF and PKU-SafeRLHF likely reflects domain mismatch: these datasets emphasize safety-related comparisons, while the UltraFeedback test distribution covers broader general-purpose comparisons.
> - Margin also matters: SHP has the highest connectivity, but UltraFeedback achieves the best accuracy because it also has a substantially larger margin. Overall, the results are consistent with our theory: good performance requires both sufficient connectivity and sufficiently large margins.
> - This experiment shows that connectivity is not merely a theoretical quantity: it is computable on real preference datasets and helps explain accuracy differences that margin alone does not.
>
>
> **3. How should one think about connectivity across datasets and models?**
>
> Across datasets, connectivity is larger when the training comparisons better cover the pairwise distinctions that matter under the target evaluation distribution. Across models, expanding the hypothesis class can decrease connectivity because the infimum is taken over a richer set of functions. This is precisely why connectivity is informative: it captures the interaction between the data distribution and the model class. In practice, this motivates estimating connectivity using a tractable hypothesis class, such as linear models, as a proxy for more complex classes: if connectivity is poor even for the linear class, it will be no larger for richer classes.
>
>
> **4. Do the bounds say anything for larger models?**
>
> We agree that the current theorems are not intended as numerically tight bounds for end-to-end large language models. Their purpose is instead explanatory: to isolate the main data-dependent factors governing learning efficiency in our analysis. The dependence on the hypothesis class is part of this story: connectivity is meant to capture how difficult it is for a given model class to generalize from the observed comparisons. Extending the theory to larger deep neural settings will likely require replacing explicit hypothesis classes with implicit function classes induced by optimization dynamics. We see this as an important direction for future work, and we will say so more clearly in the revision.
>
>
> **5. What does the KL-projection perspective imply ?**
>
> The KL-projection view clarifies that, under misspecification, discriminative BT does not in general recover the true CPRD; instead it recovers the closest approximation within the BT family at the preference level. Since our characterization shows that the BT-representable CPRDs are exactly those compatible with positive–negative conditional independence, this gives a concrete interpretation of BT suitability: one can test the conditional-independence condition, and if it is largely violated, our theory suggests either not using BT to estimate preferences or reconsidering the data collection pipeline.

---

> > ### Author Rebuttal · Reviewer_Fxxf · 2026-04-03
> >
> > Thank you for the response. The additional experimental results were helpful in showing that, despite the bounds not being tight, connectivity does have a meaningful influence on model performance. This strengthens the sample complexity analysis. The additional clarifications on the practical impact, KL perspective, and connectivity are helpful, and have addressed my concerns. Including these in the revision would strengthen the paper. However, as noted by the authors, the paper is a first step, and I also think that while the results are clear and novel, the impact remains limited. I will raise my score to a 4.

---

### Official Review · Reviewer_84Qt · 2026-03-11

**Soundness:** 3
**Presentation:** 4
**Significance:** 2
**Originality:** 3
**Overall Recommendation:** 5
**Confidence:** 3

**Summary:**

This paper proposes a framework that aims to increase understanding of preference learning from a data-centric perspective, where it is found that the connectivity and margin are the key factors for the theoretical accuracy bound. Furthermore, it introduces the conditional preference distribution (CPRD). Given a context and a pair of responses, the CPRD gives the probability that one is preferred over the other. This is defined purely from data generation, without an assumption of a generative model such as BT.

**Compliance With Llm Reviewing Policy:**

Affirmed.

**Final Justification:**

This is in my opinion a good paper, and the rebuttals suggest that the final version will address the remaining issues raised by the reviewers.

**Key Questions For Authors:**

-

**Limitations:**

Yes

**Strengths And Weaknesses:**

STRENGHTS:

This paper is strong conceptually. While literature often assumes data comes from a Bradley-Terry process, this paper actually starts from a triplet distribution (x, y+, y-) and asks what information it actually contains. Their conditional independence characterization is insightful. Theorem 4.4 is written well and gives reader a concrete, verifiable condition rather than an abstract/algebraic one. The paper separates margin and connectivity as governing factors well. The accuracy bound in Theorem 6.8 clearly decomposes into a margin term and a connectivity term, making it easy to reason about which factor is the bottleneck in a given setting. The paper is honest about its limitations.

WEAKNESSES:

The experimental setting is limited and its relevance to real-world applications is questionable. With only 16 items in 128 dimensions and a neural network generating the ground-truth scores, the setup might not be representative of actual human preference data, which tends to be noisy, inconsistent, and often not transitive. That the method works in this controlled synthetic environment does not prove that it would transfer to LLM alignment, where annotator behaviour can be much more complex. Furthermore, as the paper itself acknowledges in the discussion: the target score function is implicit, so you can't compute or increase margin in practice and although connectivity is computable, strategies for improving it remain open and no direction on how to improve them is given. This means someone reading this paper still doesn’t know what to change about their data collection pipeline, which limits the paper’s practical impact.

---

> ### Author Rebuttal · Authors · 2026-03-29
>
> We appreciate the positive assessment of the conceptual contribution, and we agree that the original submission did not make the practical implication for data collection sufficiently concrete.
>
> **1. What to change about their data collection pipeline ?**
>
> While our paper does not yet provide a full procedure for improving the preference data generation, it identifies the key quantities to try to optimize for, namely margin and connectivity, as they impact statistical efficiency.   The main practical implication of this result is that, among data collection strategies that are reasonable for the target task, one should prefer triplet distributions with better connectivity relative to the intended evaluation distribution. In this rebuttal, we also provide additional experiments on real preference dataset to evaluate this insight (see response to reviewer Fxxf, point 2).
>
> Our characterization of when the BT model is suitable for representing preference data (positive–negative conditional independence), provides a useful heuristic for assessing whether BT is a reasonable modeling assumption. If the chosen and rejected responses are strongly dependent, BT may not be the right model. In principle, this can also be probed using conditional-independence tests such as Fisher’s z-test: if the condition is substantially violated, our theory suggests either avoiding BT for preference estimation or reconsidering the data collection pipeline.
>
> We view these results as a first step toward more actionable methods for designing preference data.
>
>
>
>
> **2.Limited experiment**
>
> We agree that success in the synthetic setting does not by itself imply transfer to end-to-end LLM alignment. Since our contributions are theoretical, the role of the experiments is to provide controlled validation of the theoretical mechanism, rather than broad empirical coverage. This is why we use a synthetic setup where we can control the data-generating process, model architecture, and evaluation distribution. We will revise the paper to make this scope explicit.

---

> > ### Author Rebuttal · Reviewer_84Qt · 2026-04-03
> >
> > Thank you for your response, I am happy to stick to my original rating.

---

### Official Review · Reviewer_oHz7 · 2026-03-13

**Soundness:** 4
**Presentation:** 3
**Significance:** 3
**Originality:** 4
**Overall Recommendation:** 5
**Confidence:** 4

**Summary:**

This work gives sufficient conditions under which the Bradley-Terry (BT) model is able to recover a preference distribution from comparison based data, and gives sample complexity bounds for the case where the preference distribution is realizable by a BT model. For the realizable setting, they give a lower bound on the accuracy of the empirical risk minimizer in terms of two parameters: margin of victory of winning responses, and a connectivity parameter which captures the relative sparsity/density of the sampled completion pairs. They experimentally validate the importance of choosing a sampling distribution which optimizes for connectivity.

**Compliance With Llm Reviewing Policy:**

Affirmed.

**Key Questions For Authors:**

1. How can one test if the Bradley-Terry model is misspecified for a given preference dataset?
2. Is computing the connectivity practical for more complicated models and large datasets?

**Limitations:**

Yes

**Strengths And Weaknesses:**

I found the paper to be well written and well motivated. The authors give some nice insights, expanding the picture of how to identify when the Bradley-Terry model is misspecified, and opening up an interesting discussion of how to design sampling distributions to improve statistical efficiency in the realizable setting.

On the negative side, I found the definition of connectivity in section 6 to be difficult to parse. The examples from related work were not helpful for my intuition, as I am not familiar enough with the literature for them to be familiar to me. It would be helpful if the authors could provide more explanation for how to reason about this quantity.

The paper is also light on actionable insights for practitioners. For instance, proposition 4.2 gives a necessary and sufficient condition for representability by the BT model in terms of relative winrates of response y over response y' conditional on prompt x, but it is not clear if this yields an efficient test for detecting when the BT model is misspecified on a given preference dataset.

Similarly, the experiment in section 7.4 successful demonstrates the impact of optimizing for connectivity, but it seems difficult to even identify when connectivity is the bottleneck during dataset construction.

---

> ### Author Rebuttal · Authors · 2026-03-30
>
> Thank you for the helpful comments on intuition and practical implications.
>
> **1. More intuition on connectivity degree.**
>
> To give more intuition, let’s first consider a general test distribution $Q’$ over $\mathcal{X} \times \mathcal{Y} \times \mathcal{Y}$. The connectivity degree can be written as
>
> $$\inf_{f,g \in H} \frac{E_P[ (f-g) (\Delta))^2 ]}{ E_{Q’}[ (f-g) (\Delta))^2]}$$
>
> This general quantity is a worst-case ratio between disagreement on the training and test distribution. When $Q’ = P$, this quantity is 1.
>
> In the paper, we focus on the test distribution with the product form $Q’(x,y,y’) = Q_x(x)Q(y \mid x) Q(y’ \mid x)$. Under this choice, we evaluate over all independent pairs of responses drawn from $Q(\cdot \mid x)$ rather than only over some selective pairs. This gives the quantity a connectivity interpretation: the ratio asks whether the training distribution captures enough pairwise information to control disagreement over the full response space over independent pairs.
>
> Across datasets, connectivity is larger when the training comparisons better cover the pairwise distinctions that matter under the target evaluation distribution. Across models, expanding the hypothesis class can decrease connectivity because the infimum is taken over a richer set of functions. This is precisely why connectivity is informative: it captures the interaction between the data distribution and the model class. In practice, this motivates estimating connectivity using a tractable hypothesis class, such as linear models, as a proxy for more complex classes: if connectivity is poor even for the linear class, it will be no larger for richer classes.
>
> **2. Is computing the connectivity practical for more complicated models and large datasets?**
>
> Yes, in a prominent practical regime. If we freeze the embedding and train a reward head on top, the connectivity degree becomes much easier to estimate. In particular, for linear reward heads, it reduces to the smallest eigenvalue of a covariance-type matrix in feature space (Eq. 126). We have provided an additional experiment to compute the connectivity degree on real-world data  (see response to reviewer Fxxf, point 2)
>
> **3. How can one test if the Bradley-Terry model is misspecified for a given preference dataset?**
>
> Our result reduces the misspecification question to checking whether the preference dataset satisfies the positive–negative conditional-independence structure. Depending on the setting, one could use standard conditional-independence tools, such as Fisher’s z-test in approximately Gaussian or parametric regimes, or kernel-based tests in more flexible nonparametric settings. We do not claim that this is always easy in high-dimensional practical problems, but our characterization reframes misspecification detection as a standard statistical testing problem.
>
> Intuitively, violations are more likely when the positive and negative examples are generated in a strongly dependent way. For example, suppose an LLM starts from a negative response, edits it into a better one, and then uses the edited version as the positive response. In such a pipeline, the required conditional-independence structure may be substantially violated. Our theory then suggests that the Bradley–Terry model may not be the most appropriate model for such preference data.
>
>
> **4. Actionable takeaway.**
>
> While our paper does not yet provide a full procedure for improving the preference data generation, it identifies the key quantities to try to optimize for, namely margin and connectivity, as they impact statistical efficiency.   The main modest practical implication of this result is that, among data collection strategies that are otherwise reasonable for the target task, one should prefer triplet distributions with better connectivity relative to the intended evaluation distribution. In this rebuttal we provide additional experiments on real preference dataset to evaluate this insight (see response to reviewer Fxxf, 2).
>
> As per our discussion 2), our characterization of when the BT model is suitable for representing preference data provides a useful heuristic for assessing whether BT is a reasonable modeling assumption. If the chosen and rejected responses are strongly dependent, BT may not be the right model. We view these results as a first step toward more actionable methods for designing preference data.

---

> > ### Author Rebuttal · Reviewer_oHz7 · 2026-04-04
> >
> > Thank you for the detailed response. I have no further questions.

---

### Official Review · Reviewer_JM5a · 2026-03-13

**Soundness:** 2
**Presentation:** 1
**Significance:** 2
**Originality:** 3
**Overall Recommendation:** 4
**Confidence:** 3

**Summary:**

This manuscript studies what can be recovered from pairwise comparison triplets ((x,y^+,y^-)) without assuming from the outset that the data is generated by a Bradley-Terry (BT) model. The authors define the conditional preference distribution (CPRD) as the preference information encoded by the triplet distribution, characterize when a CPRD is representable by BT, show that the standard discriminative BT objective can be interpreted as a KL projection at the preference level, and analyze sample efficiency through two factors, margin and a proposed connectivity quantity.

**Compliance With Llm Reviewing Policy:**

Affirmed.

**Final Justification:**

In the rebuttal peroid, the authors have largely addressed my concerns.

**Key Questions For Authors:**

1.	Your main motivation is misspecification, but all experiments appear BT-consistent by construction. Can you explain why the paper does not include a non-BT CPRD experiment, and what you expect the learned BT model to recover there?

2.	In the neural-network experiments, how stable is the estimated connectivity degree from the minimax optimization described in Appendix D? If the estimate is noisy or optimization-dependent, that would affect how actionable the quantity really is.

3.	In Section 5, you introduce both the generative objective (Equation 10) and the discriminative BT objective (Equation 12). Why is there no empirical comparison between them? If you have evidence that they behave differently or similarly in finite samples, that would be very useful.

**Limitations:**

Yes

**Strengths And Weaknesses:**

Strenghts:

1. The manuscript asks a meaningful question that many preference-learning works quietly sidestep: what exactly is being learned from pairwise comparison data when the model class is BT or BT-like? That is a conceptual contribution.

2.	The CPRD formalization is a useful idea. It gives the manuscript a clean target object and helps disentangle data semantics from model assumptions.

3.	Theorem 5.2 is a strong point of the paper. Recasting discriminative BT fitting as a weighted KL projection onto the BT family is a helpful interpretive result that many readers will find clarifying.

Weaknesses:

1.	The paper’s empirical evidence does not match its motivating claims. The introduction and abstract emphasize misspecification, asking what BT recovers when the model is wrong, but the experiments in Section 7 are all generated from BT-consistent distributions. This matters because the paper’s most practically interesting claim is about misspecified real data, yet the empirical section never actually probes that regime.

2.	The experimental setup is too narrow to support broad practical conclusions. On Pages 7-8, all experiments use a tiny synthetic domain with (m=16), a matched teacher-student neural architecture, and a uniform test distribution over all pairs.

---

> ### Author Rebuttal · Authors · 2026-03-30
>
> Thank you for highlighting the point on the non-BT CPRD experiment and the stability of the connectivity degree estimation.
>
> **1. Non-BT CPRD experiment.**
>
> Our theory predicts that when the true CPRD is non-BT, BT training does not recover the true CPRD itself, but rather its KL projection onto the BT family. To verify this mechanism, we construct a synthetic non-BT CPRD by perturbing a ground-truth BT CPRD in a way that breaks BT representability while preserving the same BT projection target $r^*$. We then train BT models on samples from both the original BT CPRD and the perturbed non-BT CPRD.
>
> BT CPRD
>
> | Number of Training Samples | RMSE to $r^* (×10^{-1})$ | Pairwise Probability MAE ($×10^{-2}$) |
> |---|---|---|
> | 128 | 2.422 ± 0.302 | 6.909 ± 0.843 |
> | 512 | 2.450 ± 0.267 | 6.917 ± 0.761 |
> | 2048 | 1.909 ± 0.101 | 5.354 ± 0.265 |
> | 8192 | 1.520 ± 0.118 | 4.277 ± 0.340 |
> | 32768 | 1.019 ± 0.123 | 2.879 ± 0.344 |
> | 131072 | 0.983 ± 0.138 | 2.754 ± 0.412 |
>
> Non-BT CPRD
>
> | Number of Training Samples |RMSE to $r* (×10^{-1})$ | Pairwise Probability MAE $(×10^{-2})$ |
> |---|---|---|
> | 128 | 2.534 ± 0.300 | 10.234 ± 0.402 |
> | 512 | 2.446 ± 0.301 | 10.019 ± 0.452 |
> | 2048 | 2.098 ± 0.154 | 9.211 ± 0.179 |
> | 8192 | 1.500 ± 0.091 | 8.082 ± 0.203 |
> | 32768 | 1.033 ± 0.131 | 7.202 ± 0.433 |
> | 131072 | 0.727 ± 0.128 | 6.618 ± 0.314 |
>
> As the sample size grows, the learned reward converges to the same BT target $r^*$ in both settings, consistent with the KL projection view. At the same time, the MAE with respect to the ground truth CPRD remains substantially larger in the non-BT setting, reflecting the irreducible misspecification of the true CPRD.
>
> **2. How stable is the estimated connectivity degree from the minimax optimization?**
>
> | alpha | mean | std | relative std |
> | ----: | -------: | ------: | -----: |
> | -16.0 |   0.0333 |  0.0260 |  95.9% |
> |  -8.0 |   0.2159 |  0.0461 |  33.2% |
> |  -4.0 |   0.9569 |  0.0522 |   7.3% |
> |  -2.0 |   2.2040 |  0.0292 |   1.5% |
> |  -1.0 |   3.3330 |  0.0130 |   0.4% |
> |  -0.5 |   3.9672 |  0.0003 |  0.01% |
> |   0.0 |   3.2801 |  0.0075 |   0.2% |
> |   0.5 |   2.6066 |  0.0191 |   0.8% |
> |   1.0 |   2.0599 |  0.0296 |   1.6% |
> |   2.0 |   1.2605 |  0.0267 |   2.5% |
> |   4.0 |   0.4888 |  0.0386 |   9.8% |
> |   8.0 |   0.0850 |  0.0086 |  17.8% |
> |  16.0 |   0.0060 |  0.0020 |  44.1% |
>
>
> We provide the stability results for the connectivity degree of experiment in Section 7.3. We repeated the estimation 10 times for each value of alpha then repeated this across 5 random seeds. We report the average mean, average standard deviation, and relative standard deviation (std/mean) in the table above.
> Overall, the estimate is fairly stable in the main regime of interest. For alpha in [-4, 4], the relative standard deviation is at most about 10%, and is below 3% for most values in this range. For more extreme values of alpha, the relative variation becomes larger, but this occurs precisely when the connectivity degree itself is already very small. In those cases, the absolute standard deviation remains small (for example, 0.0260 at alpha = -16 and 0.0020 at alpha = 16).
> We will add this quantitative stability discussion to the revision to clarify that, while the minimax estimate is stochastic, it is sufficiently stable to support the trends reported in Section 7.3.
>
> **3. Why is there no empirical comparison between the generative objective and the discriminative objective?**
>
> We include the generative objective primarily as a conceptual reference point. Maximum likelihood directly corresponds to minimizing KL divergence to the true triplet distribution, which gives a clear recovery guarantee. Under correct specification, generative objective recovers the true CPRD; under misspecification, it converges to the KL projection.
>
> However, our main focus is the discriminative BT objective, both because it is a common practical approach to preference learning and because its recovery behavior under misspecification is much less obvious. Our core contribution is to show that the discriminative objective also admits a precise recovery interpretation: a KL projection at the preference level. We will revise the paper to make this role of the generative objective clearer.
>
> **4. The experimental setup is too narrow to support broad practical conclusions**
>
> We view the paper as providing a principled understanding of preference learning from triplet data. Since our contributions are theoretical, the role of the experiments is to provide controlled validation of the theoretical mechanism, rather than broad empirical coverage. This is why we use a synthetic setup where we can control the data-generating process, model architecture, and evaluation distribution. We will revise the paper to make this scope explicit.
>
> For the practical takeaway, we refer to the response to reviewer oHz7, point 4. We also provide experiments on the real-world dataset and refer to reviewer Fxxf, point 2.

---

> > ### Author Rebuttal · Reviewer_JM5a · 2026-04-06
> >
> > Thanks for the detailed responses, which largely address my concerns. I will raise the ratings. Good luck.

---

### Decision · Program_Chairs · 2026-04-30

**Decision:**

Accept (regular)

**Comment:**

This paper on statistical preference learning has received four reviews, with ratings ranging from "Weak Accept" to "Accept." Notably, the rebuttal phase was very productive, allowing the authors to effectively address several concerns raised by the reviewers. For these reasons, I recommend accepting the paper and encourage the authors to incorporate some of their discussion into the revised version.